**Cite this article:** Ścigała KA, Schild C, Zettler I. 2020 Dishonesty as a signal of trustworthiness: Honesty-Humility and trustworthy dishonesty. *R. Soc. Open Sci.* **7**: 200685.

psychology

trustworthy dishonesty, trustworthiness, trust, dishonesty, Honesty-Humility, HEXACO

**Author for correspondence:**
Karolina A. Ścigała
e-mail: kas@psy.ku.dk

# Dishonesty as a signal of trustworthiness: Honesty-Humility and trustworthy dishonesty

Karolina A. Ścigała[1], Christoph Schild[1,2] and Ingo Zettler[1]

[1]Department of Psychology, University of Copenhagen, Øster Farimagsgade 2A, 1353 Copenhagen, Denmark
[2]Department of Psychology, University of Siegen, Siegen, Nordrhein-Westfalen, Germany

 KAŚ, 0000-0003-4689-0697; CS, 0000-0002-6668-5773; IZ, 0000-0001-6140-7160

Trustworthiness is a foundation of well-functioning relationships and societies, and thus often perceived as a socially normative behaviour. Correspondingly, a broad array of research found that people tend to act in a trustworthy way and signal their trustworthiness to others, and that trustworthiness is rewarded. Herein, we explore whether this motivation to behave trustworthily can have socially undesirable effects in terms of leading to dishonesty targeted at fulfilling the trustor's expectations (i.e. *trustworthy dishonesty*). Furthermore, we examine how the basic trait of Honesty-Humility, which has consistently been found to be linked to both higher honesty and trustworthiness, relates to trustworthy dishonesty, where honesty and trustworthiness are at odds. Specifically, we conducted three pre-registered studies (*N* = 7080), introducing a novel behavioural game, the *lying-trust game*, where participants had a chance to lie to act trustworthily. In two studies, we found that, when offered 'full trust', participants high in Honesty-Humility (i.e. the top 10%) engaged in trustworthy dishonesty, i.e. lied in order to avoid maximizing their own incentive at the cost of minimizing the incentive of their trustor. This pattern was not present when the trustor offered minimal trust only, as well as among participants low in Honesty-Humility (i.e. the bottom 10%).

## 1. Introduction

Interpersonal trust is fundamental for the effective functioning of social interactions as well as of society as a whole. It has been found to be related to many societal outcomes such as lower corruption perception (e.g. [1]), higher economic growth [2], and

more efficient judicial systems [3]. Among the many definitions of interpersonal trust, one of the most comprehensive ones defines it as 'a risky choice of making oneself dependent on the actions of another in a situation of uncertainty, based upon some expectation of whether the other will act in a benevolent fashion despite an opportunity to betray' ([4]; p. 251). Thus, (interpersonal) trust is based on an interaction that includes a trustor (the one that decides to trust) and a trustee (the one that may or may not behave trustworthily). Because of this interdependence, trusting can be beneficial for trustors only when trustees fulfil the trustors' expectations regarding their behaviour (i.e. behave trustworthily; [5–7]), which can be influenced by many factors such as behavioural control, power imbalance, and conflict of interest (e.g. [5,8]). For instance, one's trustworthiness could be expressed in returning a loan borrowed from a friend, remaining faithful to one's partner, or keeping a promise to take good care of one's neighbour's cat when the neighbour is away.

Given the positive connotation of trustworthiness in social interactions, multiple studies have focused on positive and prosocial factors associated with this phenomenon (e.g. [9–11]). However, although the motivation to act trustworthily is prosocial in itself, it might, in some circumstances, lead to immoral actions. For instance, several studies found that trust and trustworthiness between interaction partners are crucial in settings of bribery and corruption (e.g. [12–14]). This opposition between the social desirability of trustworthiness on one hand and its potentially antisocial consequences on the other might result in a moral conflict in situations in which behaving trustworthily requires engaging in unethical actions, such as dishonesty. This might be especially true for people who are generally predisposed to behave in a prosocial (i.e. both honest and trustworthy) manner. In line with this possibility, in the following, we explore whether people high in Honesty-Humility (a basic personality trait positively related to a range of prosocial behaviours including trustworthiness and honesty; [7,15,16]) are willing to lie in order to act in a trustworthy way (i.e. engage in *trustworthy dishonesty*). Examples of trustworthy dishonesty include (but are not limited to) a supervisor keeping illegal conduct of their trusting employee a secret, a person covering up for a friend/family member who trusts one not to reveal their moral transgressions, or a student dishonestly taking too little credit for an assignment shared with their trusting schoolmate. In other words, trustworthy dishonesty occurs in any situation in which an individual acts in a dishonest way as a mean to fulfil another individual's expectations regarding reciprocating their trust (via acting trustworthily).

## 1.1. Trustworthiness as a social norm

Several studies have found that trustworthy behaviour is perceived as being socially normative. Bicchieri *et al.* [17], for instance, found that the majority of people believe that most people would decide to punish someone who behaved in an untrustworthy way (replicated in [18]). Indeed, van den Bos *et al.* [19] found that people who behaved untrustworthily did face punishment more often than people who acted in a trustworthy way. Relatedly, people perceived as untrustworthy were found to be trusted with less money in economic games [20], to receive lower managerial pay awards [21], and to have lower positions in corporate hierarchy [10]. Given the high costs of untrustworthiness, it is not surprising that people often tend to behave in a relatively trustworthy way. For example, meta-analytical data shows that individuals return on average 37% of the endowment they are trusted with in trust games [22]. In line with this, people tend to signal their trustworthiness with a variety of means such as engaging in charitable giving [23], moralistic punishment [24,25], or uncalculating cooperation (i.e. cooperating when it is for oneself not beneficial to do so; [26]). In summary, research suggests that trustworthiness is perceived as a social norm (e.g. [17]), that is rewarded (e.g. [20]), that people tend to act in a trustworthy way (e.g. [22]), and that people signal their trustworthiness (e.g. [25]).

## 1.2. Trustworthiness as an antisocial behaviour

Despite the vast array of research pointing towards social desirability of trustworthiness, studies on corruption and bribery suggest that the motivation to act in a trustworthy way might also lead to socially undesirable behaviour including dishonesty [12,13,27–33]. Indeed, forming and maintaining successful corrupt interactions obviously requires that the involved parties behave in a trustworthy way with regard to each other (e.g. reciprocate the corrupt offer and refrain from reporting the illegal transaction(s) to authorities; [13,14,34]). For instance, Jiang *et al.* [12] found that high trust was related to more bribery in otherwise low-corruption countries. In another study, Abbink [35] used a multi-round bribery game, in which participants were either matched with a different interaction partner every round of the game (the 'strangers' condition) or had a fixed partner throughout the game (the 'partners'

condition). Participants in the 'strangers' condition engaged in less bribery than participants in the 'partners' condition, arguably because they might not have been able to develop as much trust in their interaction partner in the former condition when compared with the latter. Similarly, Weisel & Shalvi [33]; replicated by Wouda et al. [36] found that when participants had to collaboratively engage in dishonesty to increase both their own and their partner's profits, the cheating rates were substantially higher when compared with when they could individually engage in dishonesty to increase only their own or only their partner's profits. Arguably, the increased rates of dishonesty in the collaborative context, relative to the individual context, might have been caused by a motivation to behave in a trustworthy way towards one's interaction partner. In summary, the motivation to act in a trustworthy way might lead to socially undesirable behaviour (i.e. dishonesty) when expressed in the context of corruption and bribery.

## 1.3. Honesty-Humility and trustworthy dishonesty

As outlined above, trustworthiness is a socially desirable behaviour which can, in some contexts, lead to socially undesirable behaviour such as dishonesty. This opposition might create a moral conflict between the motivation to act trustworthily and the motivation to act honestly, which might be especially salient for people who are generally prosocial (i.e. motivated to act both trustworthily and honest). One of the basic personality traits that has been consistently positively related to a broad array of prosocial behaviours (including both trustworthiness and honesty, but also cooperativeness, fairness and modesty; for meta-analytical findings, see [37,38]) is Honesty-Humility from the HEXACO Model of Personality. Honesty-Humility is defined as 'the tendency to be fair and genuine in dealing with others, in the sense of cooperating with others even when one might exploit them without suffering retaliation' ([15]; p. 156). In line with this definition, Thielmann & Hilbig [4,7] found, across three studies, that people high in Honesty-Humility behaved more trustworthily than people low in Honesty-Humility (which was replicated in [39]). On the other hand, people high in Honesty-Humility also tend to behave in a more honest way than people low in Honesty-Humility (e.g. [16,38]). In summary, people high in Honesty-Humility typically behave more trustworthily and more honest than people low in Honesty-Humility.

Herein, we will examine if participants high in Honesty-Humility are willing to lie in order to fulfil their trustors' expectations regarding their trustworthiness (i.e. engage in trustworthy dishonesty). In doing so, we explore if people high in Honesty-Humility engage in other-benefitting dishonesty as a means to act trustworthily when benefitting the other is at odds with their self-interest. Please note that we focus on the type of dishonesty in which benefitting the other is at odds with benefitting oneself in order to assure that the motivation to act trustworthily is not confounded with one's self-interest (as it is the case in bribery and corruption settings in which acting trustworthily towards one's interaction partner and self-interests are typically aligned with each other; e.g. [12,33,35]). In other words, we focus on a situation with conflicting interests between the trustor and the trustee because aligning interests between them might not give the trustee an opportunity to signal their trustworthiness without signalling their self-interest as well [8,40].

### 1.3.1. Present research

In the following, we explore whether people high in Honesty-Humility engage in trustworthy dishonesty. All of the following studies were preregistered and all the data and study materials are openly available (see https://osf.io/ywgp5). However, please note that we deviated from the preregistrations in the majority of the analyses presented below. Hence, the analyses are largely exploratory (unless specified otherwise). All of the preregistered, confirmatory analyses and a detailed explanation of why we deviated from the preregistrations are available in the electronic supplementary material.

# 2. Study 1

## 2.1. Methods

### 2.1.1. Procedure and participants

The sample size consists of 1713 participants (852 in the 'full trust' condition, and 861 in the 'no trust' condition, aged from 18 to 74 years; $M_{age} = 36.35$; s.d.$_{age} = 12.50$; 1182 females, 526 males and five 'other'). The power analysis providing a rationale for this sample size can be found in the 'sample size

rationale' section of the preregistration (https://osf.io/s2k59/). We recruited participants using the online participant pool Prolific Academic (prolific.ac; [41]) and using the survey panel formr [42]. Participants were UK residents. Participants who completed the study were paid in accordance with Prolific wage requirements (approx. £5 per hour). In addition, participants had an opportunity to obtain a bonus of up to £30, which depended on the condition they were allocated to, their own decisions, and a final lottery. Participants who failed the attention checks interspersed within the questionnaires ($N = 17$) or failed the control questions before the lying-trust game ($N = 87$) were not included in the analyses.

The data collection took place at two measurement occasions. At both measurement occasions, participants were provided with basic information about the study and were asked to give consent to participate. At the first measurement occasion, participants completed the HEXACO-60 [43] and the TOSCA-3[1] [44] in a random order. The order of the items in both questionnaires was randomized. Additionally, two attention checks were interspersed within the two questionnaires (one attention check each). At the second measurement occasion, which took place one week after the first one, participants took part in the lying-trust game. Participants were presented with the instructions, the control questions, informed about the amount passed to them by the participant they were matched with (the amount depended on whether they were randomly allocated to the 'full trust' or to the 'no trust' condition), and finally, they were asked to report an outcome of a dieroll to determine the division of the obtained money. At the end of the second measurement occasion, participants were debriefed about the purpose of the study and thanked for their participation. The two measurement occasions were divided in order to avoid a potential influence of filling in the personality questionnaires (i.e. the HEXACO-60 and the TOSCA-3) on moral decision making in the lying-trust game. All participants who completed the entire study at both measurement occasions and responded correctly to the control questions, were included in the following analyses.

### 2.1.2. Measures

#### 2.1.2.1. HEXACO-60

The HEXACO-60 is a questionnaire assessing the basic personality traits from the HEXACO Model of Personality [15,43]. Participants are presented with 60 statements about themselves and people in general, and they are asked to rate the degree to which they (dis)agree with these statements using a five-point Likert scale (ranging from 1 = 'strongly disagree' to 5 = 'strongly agree'). The Honesty-Humility scale includes items such as 'I would not use flattery to get a raise or promotion at work, even if I thought it would succeed.', or 'I would never accept a bribe, even if it were very large'. To account for inattentive responding, we included an attention check, namely 'This is an attention check, please choose 1'. Although the focus of this study is primarily on Honesty-Humility, we decided to administer the whole HEXACO-60 to avoid bringing the Honesty-Humility content into the focus of participants' attention.

#### 2.1.2.2. The lying-trust game

The lying-trust game includes two players who are matched with each other (player 1 is the trustor, and player 2 is the trustee). Because in this study, we are interested in the behaviour of the trustees only, all participants were assigned to the role of players 2 (the trustees), whereas the data from players 1 (the trustors) was collected in a preceding pilot study (thus avoiding deception). First, participants were informed that players 1, who have already participated in the study, received £10 and could have chosen to pass them an amount of either £1 or £10, which then gets tripled and becomes a part of participants' (potential) earnings. Next, participants were instructed that after they receive the tripled amount (i.e. either £3 or £30), they will have a chance to divide the money between themselves and player 1 based on the (reported) outcome of an anonymous dieroll. To account for the fact that some participants might not own a die, we asked participants to roll the die online. In addition, to maintain anonymity, the dieroll took place at an external website (https://www.random.org/dice/?num=1), so that we had no access to the actual outcomes of the dieroll. Participants were informed that the higher the number they report, the more money they get, and the less money player 1 gets. On the other hand, the lower the number they report, the less money they get and the more money player 1 gets. In other words, participants could act dishonestly—either by reporting a higher number than they

---

[1]The results related to the TOSCA-3 are not relevant to this manuscript and are therefore not reported herein. All the pre-registered findings including the TOSCA-3 can be found in the electronic supplementary material.

**Table 1.** Incentive structure across three studies. (Note: P1, player 1; P2, player 2. In Studies 2 and 3, only the 'full trust' condition was administered.)

| dieroll outcome | Study 1 'full trust' condition | | 'no trust' condition | | Study 2 | | Study 3 | |
|---|---|---|---|---|---|---|---|---|
| | P1 | P2 | P1 | P2 | P1 | P2 | P1 | P2 |
| 1 | £30 | £0 | £3 | £0 | $30 | $0 | £30 | £0 |
| 2 | £24 | £6 | £2.4 | £0.6 | $24 | $6 | £24 | £6 |
| 3 | £18 | £12 | £1.8 | £1.2 | $18 | $12 | £18 | £12 |
| 4 | £12 | £18 | £1.2 | £1.8 | $12 | $18 | £12 | £18 |
| 5 | £6 | £24 | £0.6 | £2.4 | $6 | $24 | £6 | £24 |
| 6 | £0 | £30 | £0 | £3 | $0 | $30 | £0 | £30 |

actually rolled (i.e. over-report) or by reporting a lower number than they actually rolled (i.e. under-report). The exact incentive structure depending on the dieroll outcome is presented in table 1. The bonuses based on the decisions in the lying-trust game were materialized for 50 dyads involved in each study. In all studies, participants knew before they made their decisions that their decisions have consequences for themselves as well as for the participants they were matched with, and that the bonuses are based on the above-mentioned lottery.

After reading the instructions, participants were presented with three control questions, namely: (i) 'Suppose that player 1 chooses to pass £1 to you. How much money will you have, before dividing the money between you and player 1? (a) £1, (b) £3, (c) £30'; (ii) 'The higher number you report as the outcome of the dieroll, (a) the more money you receive, and the less money player 1 receives, (b) the less money you receive, and the more money player 1 receives'; and (iii) 'The lower number you report as the outcome of the dieroll, (a) the more money you receive, and the less money player 1 receives, (b) the less money you receive, and the more money player 1 receives'. If participants failed at least one of the questions, they were asked to read the instructions and answer the questions again. If they failed to answer any question for the second time, they were excluded from the study.

After successful completion of the control questions, participants were randomly assigned to either the 'full trust' or the 'no trust' condition. In the 'full trust' condition, participants were informed that player 1 decided to pass £10 to them and keep £0 for themselves, which results in a sum of £30 in their disposition. On the other hand, in the 'no trust' condition, participants were informed that player 1 decided to pass £1 to them and keep £9 for themselves, which results in a sum of £3 in their disposition. Then, participants were asked to anonymously roll an electronic die to determine the division of the obtained amount between themselves and player 1. The table with the incentives for both participants depending on the outcome of the dieroll was present on the screen when participants were reporting their dieroll outcomes.

## 2.2. Results

### 2.2.1. Trusting behaviour and dishonesty

We compared the reported outcomes of the dieroll in both conditions to the outcomes that would be expected assuming full honesty. Participants' reports in the 'full trust' condition did not differ significantly from what would be expected assuming full honesty, as indicated by the Wilcoxon signed-rank test ($M = 3.51$; s.d. $= 1.61$; $V = 184\,421$; $p = 0.700$). On the other hand, participants in the 'no trust' condition reported higher outcomes when compared with what would be expected assuming full honesty, ($M = 3.95$; s.d. $= 1.71$; $V = 238\,769$; $p < 0.001$). Furthermore, we compared the reported outcomes in both conditions using ordinal logistic regression (predicting the outcomes of the dieroll with condition). We found that participants in the 'no trust' condition reported significantly higher outcomes than participants in the 'full trust' condition ($b = 0.49$, s.e. $= 0.09$; $z = 5.74$; $p < 0.001$; odds ratio (OR) $= 1.64$; 95% confidence interval (CI) $= (1.38; 1.94)$; figure 1).

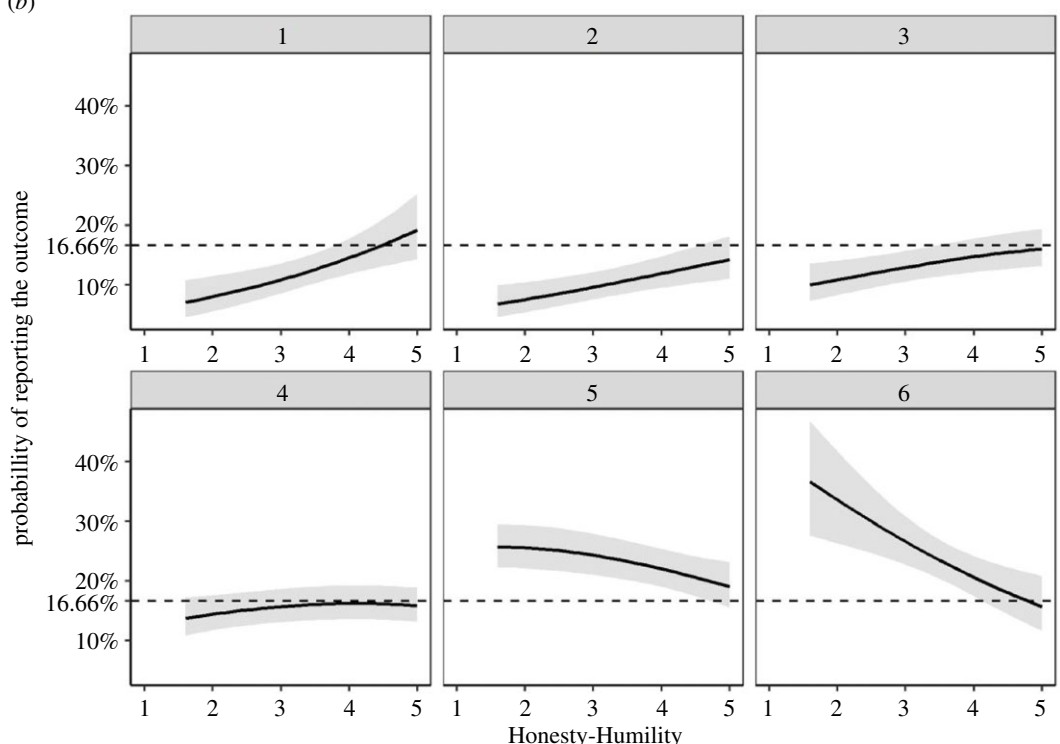

**Figure 1.** (*a,b*) Probabilities of reporting dieroll outcomes in the full and 'no trust' conditions by Honesty-Humility in Study 1. The dotted horizontal lines indicate the expected probability of reporting each number assuming full honesty. The grey ribbon illustrates 95% CIs. *N* = 852 ('full trust' condition); *N* = 861 ('no trust' condition).

### 2.2.2. Trustworthy dishonesty and Honesty-Humility

We found that Honesty-Humility was negatively related to the reported outcomes of the dieroll in both the 'full trust' (*b* = −0.27, s.e. = 0.09; *z* = −2.92; *p* = 0.002; OR = 0.76; 95% CI = (0.64; 0.91)) and the 'no trust'

($b = -0.33$, s.e. = 0.09; $z = -3.78$; $p < 0.001$; $OR = 0.72$; 95% CI = (0.60; 0.85)) conditions. To gain a better understanding of the relationship between Honesty-Humility and trustworthy dishonesty, we plotted the relations between Honesty-Humility and the probability of reporting each of the dieroll outcomes together with an indication of the probability of reporting each of the outcomes assuming full honesty (i.e. the honesty benchmark; figure 1$a$ and $b$). As can be seen in figure 1$a$, the entire 95% CI of the probability of reporting the income-maximizing outcome (six) is below the honesty benchmark for people high in Honesty-Humility, whereas the 95% CI of the probability of reporting the equalizing outcomes (three and four) falls entirely above the honesty benchmark for people high in Honesty-Humility. This suggests that some people high in Honesty-Humility may have under-reported the income-maximizing outcome, and may have chosen to report one of the equalizing outcomes instead, implying that they acted dishonestly to benefit their partner. Furthermore, such a pattern was not present in the 'no trust' condition (figure 1$b$), suggesting that participants high in Honesty-Humility engaged in dishonest behaviour as a means to reciprocate previous trusting behaviour of their partner.

Because the method used above (i.e. the plots) does not allow us to calculate the proportion of under-reporting/over-reporting dishonest individuals, and is purely based on figure interpretation, we employed another analytical procedure following an approach suggested by Moshagen & Hilbig [45]. This procedure lets us estimate the proportions of participants who engaged in dishonest under- and over-reporting. Using this procedure is necessary because the dieroll outcomes obtained in the lying-trust game are anonymous and hence we cannot determine which participants actually rolled the numbers they reported and which participants did not (this kind of anonymity is necessary to give participants adequate conditions for cheating; [46]). In other words, the number of participants who indicated that they rolled a given number reflects both the number of participants who actually rolled the number and the number of participants who dishonestly misreported the number. Hence, to obtain an accurate proportion of dishonest individuals, we have to take into account the objective probability of rolling each number in one dieroll (which, in our study, equals 1/6; [45]). Please note that this method has been shown to outperform traditional analytical approaches ([45]) and is increasingly used in studies on dishonesty (e.g. [16,47,48]). We estimated the proportion of over-reporting individuals ($d$) as $d = (p(\text{win}) - p)/(1 - p)$ in which $p(\text{win}) = d + (1 - d) * p$. That is, the proportion of winners responses ($p(\text{win})$) is a function of the proportions of honest and dishonest individuals and the baseline probability $p$ (i.e. the objective chance of winning). Furthermore, we used an analogical approach to estimate the proportion of under-reporting dishonest individuals by slightly adjusting our approach such that $p(\text{loose}) = u + (1 - u) * p$, where $u$ refers the proportion of under-reporting individuals. Solving for $u$ thus yields $u = (p(\text{loose}) - p)/(1 - p)$. In our study, $p$ is 1/6, because this is the objective probability to roll a given number in one dieroll. To test whether $d/u$ are significantly different from 0, we performed likelihood-ratio tests using MULTITREE ([49]; the model equations are shown in the Open Science Framework; link above). Because there is currently no available procedure to estimate a linear relation between a continuous variable (in our case, Honesty-Humility) and under-reporting, in the following we decided to calculate both probabilities for the participants with the top and bottom, respectively, 10% scores in Honesty-Humility ($N = 170$ in the 'full trust' condition, and $N = 172$ in the 'no trust' condition). The detailed reasoning behind the choice of the 10% cut-off is outlined in the electronic supplementary material (pp. 1–6). In brief, based on visual examination of the data, we concluded that—in Studies 1 and 3—participants with the top 10% scores in Honesty-Humility consistently under-reported the outcome that maximized their own income and minimized their partner's income. In addition, because the current methods (i.e. MULTITREE, RRreg; [49,50]) allow to obtain a valid $p$-value for over-reporting, but not for under-reporting, the following results are based on CI interpretation—i.e. estimates which CIs include zero are interpreted as insignificant, whereas estimates which CIs do not include zero are interpreted as significant.

The results show that participants high in Honesty-Humility (i.e. the top 10%) over-reported fours (an equalizing outcome; $P_o = 0.17$; 95% CI = (0.05; 0.29)) and under-reported sixes (an income maximizing outcome; $P_u = 0.57$; 95% CI = (0.22; 0.86)) in the 'full trust' condition (for the remaining outcomes; table 2). This suggests that some of the participants high in Honesty-Humility who observed the income-maximizing outcome may have chosen to report the equalizing outcome instead. Furthermore, a similar pattern was not present in the 'no trust' condition. Therein, participants high in Honesty-Humility were honest about both the equalizing and the income-maximizing outcomes ($P_o = 0.02$; 95% CI = (0; 0.11), and $P_u = 0.10$; 95% CI = (0; 0.44), respectively).

In the following comparisons (between participants high and low in Honesty-Humility, and between conditions), we will focus on the proportion of participants dishonestly under-reporting the income maximizing outcome, which is the only outcome that can interpreted as a pure indication of

**Table 2.** Proportions of over- and under-reporting dishonest individuals (Study 1; 'full trust' condition), $N = 170$. (Study 1; 'no trust' condition), $N = 172$. (Note: HH, Honesty-Humility; 95% CIs of the values in bold do not include zero.)

| dieroll | proportion of over-reporting dishonest individuals | | proportion of under-reporting dishonest individuals | |
|---|---|---|---|---|
| | high HH | low HH | high HH | low HH |
| 'full trust' condition | | | | |
| 1 | 0.05 (0; 0.14) | 0.02 (0; 0.10) | 0.10 (0; 0.44) | 0.24 (0; 0.65) |
| 2 | 0.02 (0; 0.10) | 0.02 (0; 0.10) | 0.18 (0; 0.58) | 0.38 (0; 0.72) |
| 3 | 0.02 (0; 0.10) | 0.02 (0; 0.10) | 0.24 (0; 0.65) | 0.14 (0; 0.51) |
| 4 | **0.17 (0.05; 0.29)** | 0.11 (0; 0.24) | 0.09 (0; 0.44) | 0.10 (0; 0.44) |
| 5 | 0.02 (0; 0.10) | 0.05 (0; 0.15) | 0.14 (0; 0.51) | 0.09 (0; 0.44) |
| 6 | 0.02 (0; 0.10) | 0.02 (0; 0.11) | **0.57 (0.22; 0.86)** | 0.14 (0; 0.51) |
| 'no trust' condition | | | | |
| 1 | 0.04 (0; 0.12) | 0.02 (0; 0.11) | 0.10 (0; 0.44) | **0.59 (0.23; 0.86)** |
| 2 | 0.02 (0; 0.11) | 0.02 (0; 0.09) | **0.44 (0.09; 0.79)** | **0.45 (0.02; 0.79)** |
| 3 | 0.02 (0; 0.09) | 0.02 (0; 0.11) | 0.14 (0; 0.51) | 0.11 (0; 0.44) |
| 4 | 0.02 (0; 0.11) | 0.02 (0; 0.11) | 0.19 (0; 0.58) | 0.37 (0; 0.72) |
| 5 | 0.08 (0; 0.18) | 0.05 (0; 0.16) | 0.10 (0; 0.44) | 0.09 (0; 0.44) |
| 6 | 0.05 (0; 0.13) | **0.23 (0.11; 0.36)** | 0.10 (0; 0.44) | 0.11 (0; 0.44) |

trustworthy dishonesty on its own. On the other hand, over-reporting the equalizing outcomes can also indicate self-interested dishonesty if accompanied by actually obtaining a lower outcome. The proportion of participants dishonestly under-reporting the income-maximizing outcome among the participants high in Honesty-Humility was higher in the 'full trust' condition than in the 'no trust' condition ($\Delta G^2(1) = 6.15$; $p = 0.019$; $\omega = 0.13$). Hence, some participants high in Honesty-Humility chose to dishonestly increase the profit of their partner by under-reporting the income-maximizing outcome, as a means to reciprocate the previous trusting behaviour via dishonesty. Importantly, such a behavioural pattern was not the case for participants low in Honesty-Humility (i.e. the bottom 10%) who did not under-report the income maximizing outcome ($P_u = 0.14$; 95% CI = (0; 0.51)) in the 'full trust' condition. However, we did not find a significant difference between participants high and low in Honesty-Humility in terms of the proportion of individuals dishonestly under-reporting the income-maximizing outcome in the 'full trust' condition ($\Delta G^2(1) = 2.97$, $p = 0.074$, $\omega = 0.12$). Thus, the results do not provide an entirely conclusive picture of the relation in question (especially in the light of a rather low sample size to test for such a difference). As a remedy, in the following, we present two replicationship studies focusing on the relation between Honesty-Humility and trustworthy dishonesty in the 'full trust' condition.

Please note that the detailed results for other cut-offs (namely, 25% and a median split) are reported in the electronic supplementary material. In brief, similarly to participants in the top 10% Honesty-Humility, both participants in the top 25% Honesty-Humility and with Honesty-Humility values higher than the median under-reported the income-maximizing outcome (six) in the 'full trust' condition, but not in the 'no trust' condition, and over-reported the equalizing outcome (four) in the 'full trust' condition, but not in the 'no trust' condition.

# 3. Study 2

Study 2 constitutes a partial replication of Study 1 conducted in order to test the replicability of the relationship between Honesty-Humility and trustworthy dishonesty in a larger sample and a different participant panel.

**Table 3.** Proportions of over- and under-reporting dishonest individuals (Study 2). (Note: HH, Honesty-Humility; 95% CIs of the values in bold do not include zero; $N = 444$.)

| dieroll | proportion of over-reporting dishonest individuals | | proportion of under-reporting dishonest individuals | |
|---|---|---|---|---|
| | high HH | low HH | high HH | low HH |
| 1 | 0.01 (0; 0.06) | 0.01 (0; 0.06) | **0.48 (0.24; 0.68)** | **0.54 (0.32; 0.73)** |
| 2 | 0.01 (0; 0.06) | 0.01 (0; 0.06) | **0.46 (0.24; 0.68)** | **0.68 (0.49; 0.86)** |
| 3 | 0.05 (0; 0.11) | 0.01 (0; 0.05) | 0.06 (0; 0.27) | **0.32 (0.05; 0.57)** |
| 4 | **0.06 (0.00003; 0.13)** | 0.01 (0; 0.06) | 0.06 (0; 0.30) | 0.08 (0; 0.32) |
| 5 | 0.06 (0; 0.12) | **0.10 (0.03; 0.17)** | 0.06 (0; 0.27) | 0.06 (0; 0.30) |
| 6 | 0.03 (0; 0.09) | **0.22 (0.15; 0.30)** | 0.06 (0; 0.30) | 0.06 (0; 0.30) |

## 3.1. Methods

### 3.1.1. Procedure and participants

The sample includes 2230 participants (aged from 18 to 78 years, $M_{age} = 38.29$; s.d.$_{age} = 12.27$; 1149 males, 1072 females and nine 'other'). The sample size rationale with a corresponding power analysis is available in the 'Sample size rationale' section in the preregistration (https://osf.io/dtk93). The participants were recruited using the participant panel Mechanical Turk (mturk.com; [51]). All participants were US residents and were above 18 years old. They were paid approximately US$ 7 per hour, and had a chance to win a bonus incentive of up to US$ 30 (depending on the condition, their decisions and a lottery). Participants who failed the attention checks interspersed within the questionnaires ($N = 73$) or failed the control questions before the lying-trust game ($N = 135$) were not included in the analyses.

The data collection was identical to Study 1 with two exceptions: (i) participants were not asked to fill in the TOSCA-3, and (ii) only the 'full trust' condition of the lying-trust game was included. For details on these measures, see the 'Measures' section of Study 1.

## 3.2. Results

Overall, the outcomes of the dieroll were significantly higher ($M = 4.13$; s.d. $= 1.54$) than what would be expected assuming full honesty ($V = 1781996$, $p < 0.001$). Furthermore, we found that Honesty-Humility was negatively related to the reported outcomes of the dieroll ($b = -0.27$, s.e. $= 0.05$; $z = -5.49$; $p < 0.001$; OR $= 0.76$; 95% CI $= (0.69; 0.84)$). The next part of the analyses focuses on participants in the top and the bottom 10% in Honesty-Humility ($N = 444$).

In line with Study 1, we found that participants high in Honesty-Humility (i.e. the top 10%;) over-reported the equalizing outcome (i.e. four; $P_o = 0.06$; 95% CI $= (0.00003; 0.13)$). In contrast to Study 1, we did not find any indication of participants high in Honesty-Humility under-reporting the income-maximizing outcome ($P_u = 0.06$; 95% CI $= (0; 0.30)$). On the contrary, in this study participants high in Honesty-Humility under-reported ones and twos (outcomes resulting in the lowest profit for oneself; $P_u = 0.48$; 95% CI $= (0.24; 0.68)$ and $P_u = 0.46$; 95% CI $= (0.24; 0.68)$, respectively; for the remaining outcomes; table 3). These results are also supported by the interpretation of the 95% CIs relative to the honesty benchmark (figure 2). Please note that the detailed results for other cut-offs (namely, 25% and a median split) are reported in the electronic supplementary material. In brief, participants high in Honesty-Humility (both in the top 25% and with values higher than the median) over-reported the equalizing outcome (four) and under-reported the income-minimizing outcomes (ones and twos) similarly to participants in the top 10%.

From a general perspective, it should be noted that the mean of reported numbers in this study ($M = 4.13$; s.d. $= 1.54$) was significantly higher than the mean of reported numbers in the 'full trust' condition in Study 1 ($M = 3.51$; s.d. $= 1.61$; $W = 1\,157\,945$; $p < 0.001$). This indicates that participants in this study overall cheated substantially more (in terms of selfish dishonesty) than participants in Study 1, suggesting a plausible reason why the results from this study are not (fully) in line with what was observed in Study 1. Specifically, because our indicator of under-reporting is probabilistic (i.e. we do not see under-reporting on an individual level, but rather infer it based on the probability

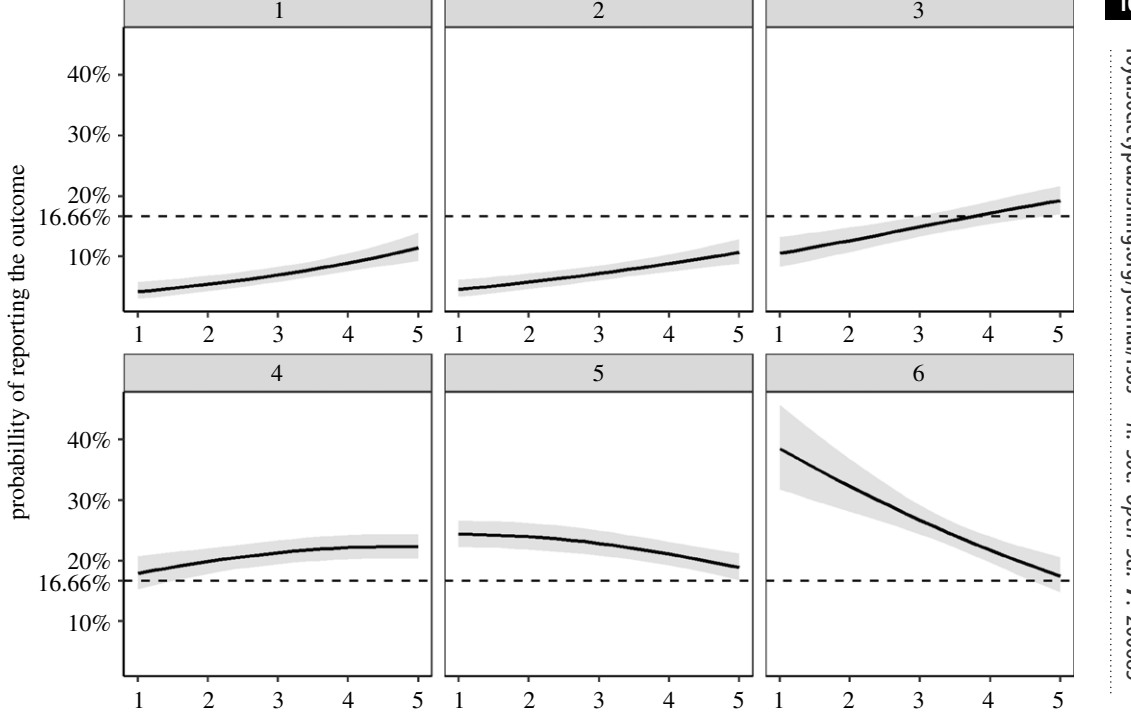

**Figure 2.** Probabilities of reporting dieroll outcomes by Honesty-Humility in Study 2. The dotted horizontal line indicates the expected probability of reporting each number assuming full honesty. The grey ribbon illustrates 95% CIs. $N = 2230$.

distribution), the few individuals that might have under-reported the income-maximizing outcome might have been 'outweighted' by the overall high proportion of dishonest individuals in terms of over-reporting. To tackle this limitation, we conducted a third study, again on Prolific Academic (but recruiting a larger sample than in Study 1), where (as indicated by Study 1) the overall proportion of dishonest individuals was supposed to be lower (which was also found) than on Mechanical Turk.

## 4. Study 3

Study 3 constitutes a partial replication of Study 1 and a direct replication of Study 2 on the same (in case of Study 1) and a different (in case of Study 2) panel platform, using a larger sample size (in case of both studies). Study 3 was conducted in order to shed light on the inconclusive findings within and across Studies 1 and 2.

### 4.1. Procedure and participants

The sample consists of 3137 participants (aged from 18 to 86 years, $M_{age} = 36.23$; s.d.$_{age} = 12.14$; 2120 females, 999 males and 18 'other'). The power analysis can be found in the preregistration (the 'Sample size rationale' section; https://osf.io/x8mz3). The sample was obtained using Prolific Academic (prolific.ac; [41]). Similarly to Study 1, all participants were at least 18 years old, were UK residents, were paid according to Prolific Academic wage standards (approx. £5 per hour), and had a chance to win a bonus of up to £30. Participants who failed the attention checks interspersed within the questionnaires ($N = 21$) or failed the control questions before the lying-trust game ($N = 130$) were not included in the analyses. The data collection was identical to Study 2, with the exception of the panel provider.

### 4.2. Results

The mean of the reported numbers ($M = 3.76$; s.d. $= 1.58$) was significantly higher than what would be expected assuming full honesty ($V = 2\,913\,383$; $p < 0.001$). In line with Studies 1 and 2, Honesty-Humility was negatively related to the reported outcomes of the dieroll ($b = -0.19$, s.e. $= 0.05$; $z = -4.15$;

**Table 4.** Proportions of over- and under-reporting dishonest individuals (Study 3). (Note: HH, Honesty-Humility; 95% CIs of the values in bold do not include zero; $N = 628$.)

| dieroll | proportion of over-reporting dishonest individuals | | proportion of under-reporting dishonest individuals | |
| --- | --- | --- | --- | --- |
| | high HH | low HH | high HH | low HH |
| 1 | 0.01 (0; 0.05) | 0.01 (0; 0.05) | 0.21 (0; 0.45) | **0.54 (0.35; 0.71)** |
| 2 | 0.01 (0; 0.05) | 0.01 (0; 0.06) | **0.37 (0.16; 0.58)** | **0.30 (0.08; 0.52)** |
| 3 | **0.07 (0.01; 0.12)** | 0.02 (0; 0.06) | 0.05 (0; 0.24) | 0.05 (0; 0.25) |
| 4 | **0.06 (0.01; 0.11)** | **0.07 (0.01; 0.13)** | 0.05 (0; 0.24) | 0.05 (0; 0.24) |
| 5 | 0.05 (0; 0.10) | 0.03 (0; 0.08) | 0.05 (0; 0.24) | 0.05 (0; 0.25) |
| 6 | 0.01 (0; 0.05) | **0.06 (0.003; 0.11)** | **0.28 (0.04; 0.48)** | 0.05 (0; 0.22) |

$p < 0.001$; OR = 0.82; 95% CI = (0.75; 0.90)). The next part of the analyses focuses on participants in the top and the bottom 10% in Honesty-Humility ($N = 628$).

Participants high in Honesty-Humility (i.e. the top 10%) over-reported both equalizing outcomes (i.e. threes; $P_o = 0.07$; 95% CI = (0.01; 0.12), and fours; $P_o = 0.06$; 95% CI = (0.01; 0.11)) and under-reported the income-maximizing outcome (six; $P_u = 0.28$; 95% CI = (0.04; 0.48); for the remaining outcomes, table 4). The same conclusions can be drawn from interpreting the 95% CIs of the probability of reporting the dieroll outcomes relative to the honesty benchmark (figure 3). Importantly, participants high in Honesty-Humility under-reported the income-maximizing outcome to a larger degree than people low in Honesty-Humility ($P_u = 0.05$; 95% CI = (0; 0.22); $\Delta G^2(1) = 10.35$; $p = 0.006$; $\omega = 0.07$).

The results suggest that some participants high in Honesty-Humility who observed the income-maximizing outcome might have decided to report the equalizing outcomes instead. Consequently, people high in Honesty-Humility when compared with people low in Honesty-Humility showed higher levels of trustworthy dishonesty. Importantly, however, some participants high in Honesty-Humility also under-reported an income-minimizing outcome (two; $P_u = 0.37$; 95% CI = (0.16; 0.58)) which suggests that some participants high in Honesty-Humility who observed the income-minimizing outcome might have chosen to report an equalizing outcome instead. This suggests that some of the participants high in Honesty-Humility engaged in self-interested dishonesty, i.e. lying for their own interest.

Please note that the detailed results for other cut-offs (namely, 25% and a median split) are reported in the electronic supplementary material. In brief, similarly to participants in the top 10%, both participants in the top 25% and with values higher than the median over-reported the equalizing outcomes (threes and fours). On the other hand, neither participants in the top 25%, nor with values higher than the median under-reported the income-maximizing outcome (six).

# 5. General discussion

Trustworthiness is a trait that enables the effective functioning of individual relationships and societies at large. The motivation to behave in a trustworthy way might, however, lead to committing immoral actions, such as engaging in trustworthy dishonesty, that is, behaving dishonestly in order to fulfil the expectations of one's trustor. In Studies 1 and 3, we found that participants high in Honesty-Humility (in the top 10%) engaged in trustworthy dishonesty. Specifically, we found that, when fully trusted, some participants high in Honesty-Humility chose to lie about not obtaining the outcome that would maximize their own outcome at the cost of the trustor. In addition, when fully trusted, some participants high in Honesty-Humility lied that they obtained the outcome that would equalize the incentives between themselves and their trustor. Such a pattern, however, was not observed among participants low in Honesty-Humility (bottom 10%; across all three studies), and was not observed when the trustor did not offer their trust to the participants (in Study 1).

By introducing the concept of trustworthy dishonesty and investigating its link to Honesty-Humility, our investigation is indeed one of a very few that captures a situational context in which people high in Honesty-Humility act in an arguably unethical way (see also, [52] or experiment 3 [53]). Importantly, we found that our conclusions can only be attributed to participants with the top 10% scores in Honesty-

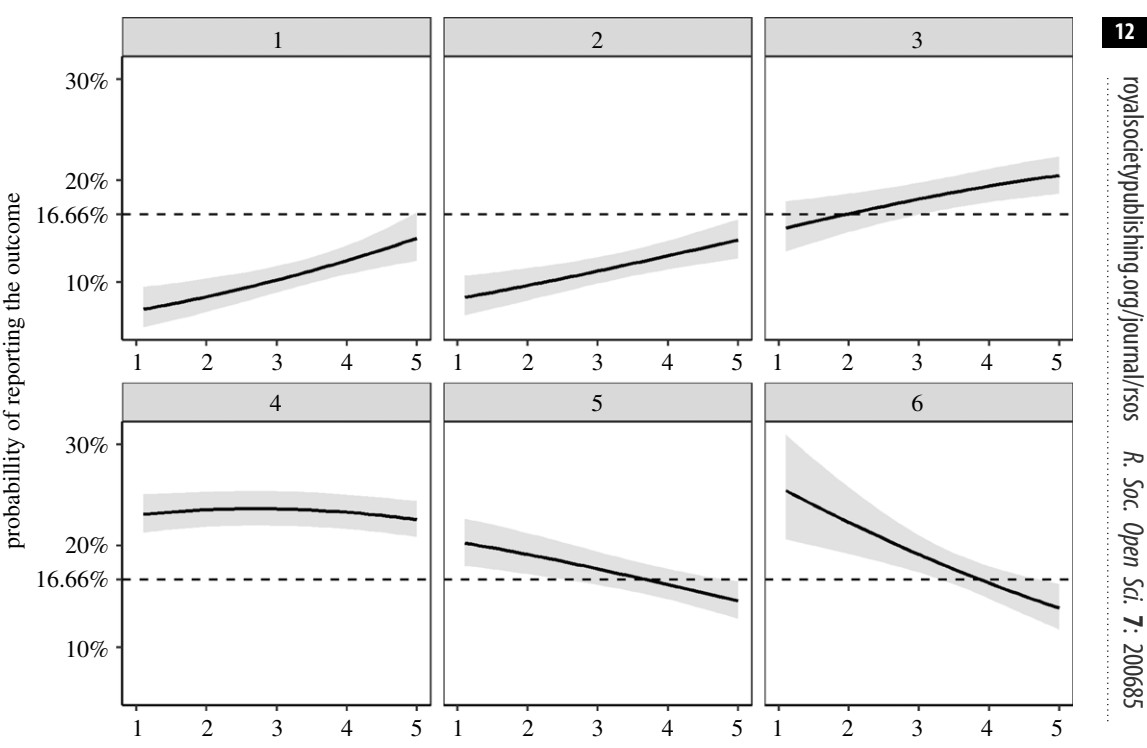

**Figure 3.** Probabilities of reporting dieroll outcomes by Honesty-Humility in Study 3. The dotted horizontal line indicates the expected probability of reporting each number assuming full honesty. The grey ribbon illustrates 95% CIs. $N = 3137$.

Humility, pointing towards potential implications for these and other studies. Specifically, it appears possible that some of the relationships between Honesty-Humility (or other personality traits) and behavioural outcomes that have been observed in the literature might be driven by participants with the very highest/lowest scores only.

More generally, our conclusions should be treated as preliminary, because there are several alternative explanations of our findings. First of all, it should be noted that inequity-aversion (e.g. [54,55]) might have played a role in in our findings because it has already been found to play a role in trust games [55]. Specifically, participants who engaged in other-benefitting dishonesty in the 'full trust' condition increased equity between themselves and their partner (because their trustors did not keep anything from their initial endowment), whereas participants who engaged in other-benefitting dishonesty in the 'no trust' condition decreased equity between themselves and their partner (because their trustors kept 90% of their initial endowment). Consequently, it might be that it was inequity-aversion rather than trustworthiness (or a mixture of both) that motivated participants to engage in other-benefitting dishonesty to a larger degree in the 'full trust' rather than in the 'no trust' condition. Second, Leib et al. [56] found that people engaged in dishonest helping to a larger degree after a fair when compared with an unfair treatment. In line with this, it is possible that participants in the 'no trust' condition engaged in less other-benefitting dishonesty when compared with participants in the 'full trust' condition because participants in the 'no trust' condition perceived the amount of money they received from the trustee as unfair. Third, our results might be attributed to the characteristics of the task. Specifically, participants high in Honesty-Humility might have chosen to engage in trustworthy dishonesty (in our Studies 1 and 3) because being asked to roll a die on an anonymous, external website might have been interpreted as an implicit license to lie (e.g. [57]) and hence as ethically acceptable in the context of the task. Finally, participants in the top 10% in Honesty-Humility were older and included more women than the remaining 90% (for details, see the electronic supplementary material, p. 20), which might have influenced the results (based on meta-analytic results, both older participants and women generally cheat less than younger participants and men, respectively; [58]).

Furthermore, it should be noted that the findings were not fully consistent across the three studies. We observed trustworthy dishonesty among people high in Honesty-Humility on Prolific Academic, but the findings did not fully replicate on Mechanical Turk. Specifically, although we did observe in

Study 2, too, that people high in Honesty-Humility cheated in terms of over-reporting the equalizing outcome, we did not observe cheating in terms of under-reporting the outcome maximizing their own incentive at the cost of the trustor. It should be noted, however, that the overall proportion of dishonest individuals (in terms of cheating to increase one's own profit) were substantially higher on Mechanical Turk than on Prolific Academic. Because our indicators of dishonesty are probabilistic (i.e. we are not able to infer if someone cheated on the individual level, but rather on the group level only), we might not have been able to statistically observe the few individuals that might have chosen to under-report the income-maximizing outcome on Mechanical Turk. In addition, it should be noted that we obtained a significant difference between participants high and low in Honesty-Humility in terms of under-reporting of the income-maximizing outcome in Study 3 only. Although there are plausible explanations for why that was the case—i.e. a small sample size in Study 1 and no under-reporting of the income-maximizing outcome in Study 2—the conclusions should be treated preliminarily. Relatedly, the majority of the conduced analyses were exploratory—especially with regard to which low and top percentages were applied to separate people low and high in Honesty-Humility—pointing at a necessity to conduct pre-registered, confirmatory studies on this topic, based on the effect sizes and patterns found across our studies. Finally, the lower bounds of the CIs for several of our significant findings were only slightly higher than zero (which would indicate an insignificant finding), further pointing at a necessity of future replication of our findings.

Overall, these studies provide first insights into the phenomena of trustworthy dishonesty, and into the relationship between Honesty-Humility and two moral obligations—to be trustworthy and to be honest—in a context in which these obligations are at odds. Our findings suggest that for participants with the highest scores in Honesty-Humility (i.e. the top 10%), the motivation to be trustworthy can override the motivation to be honest. We hope that future studies will test the robustness of our findings. In addition, we hope that future studies will focus on the relationships between other personality traits and behaviours in situations in which various (moral) obligations are at odds, shedding some light on what behavioural tendencies are fundamental for different personality traits.

Ethics. The ethical approval (IP-IRB / 22032019) was given by the Institutional Ethical Review Board, University of Copenhagen, Department of Psychology. All participants gave informed consent to participate in the study.

Data accessibility. All the analyses scripts and data are openly available (see; Ścigała, Schild, Zettler, 2020; https://osf.io/ywgp5/).

Authors' contributions. K.A.Ś., C.S. and I.Z. designed research, K.A.Ś., C.S. and I.Z. performed research, K.A.Ś. and C.S. analysed data and K.A.Ś., C.S. and I.Z. wrote the paper.

Competing interests. We declare we have no competing interests.

Funding. This investigation was funded by grants from the Carlsberg Foundation (CF16-0444) as well as the Independent Research Fund Denmark (7024-00057B) to the last author.

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
