## [Reviewer comments · Royal Society Open Science]

Review History

RSOS-200685.R0 (Original submission)

Review form: Reviewer 1

Is the manuscript scientifically sound in its present form?

Yes

Are the interpretations and conclusions justified by the results?

No

Is the language acceptable?

Yes

Do you have any ethical concerns with this paper?

Yes

Have you any concerns about statistical analyses in this paper?

Yes

Recommendation?

Major revision is needed (please make suggestions in comments)

Comments to the Author(s)

The authors investigate a phenomenon termed 'trustworthy dishonesty', i.e. dishonesty committed in service of honoring another person's trust. They test whether the personality trait honesty-humility (HH) is predictive of such trustworthy dishonesty, which would inform the interpretation of this personality trait in a situation where trustworthiness and honesty are at odds. In study 1, they find that people who score high on HH tend to underreport payoff-maximizing events (rolling a 6) and overreport equalizing events (rolling a 3 or 4), thus exhibiting trustworthy dishonesty, more so than people with low HH. They replicate this finding in study 3 but not study 2 and note interesting differences in behavior between Prolific and MTurk samples.

In my view, the authors present an elegant experiment with interesting results. There are many papers showing that people cheat on die-rolling tasks for their own benefit, but this work shows that some people are willing to cheat even for the benefit of others. I appreciate the open science-mindedness of the authors, and I do not have major objections to the analytical approach (although I do have several questions/comments below).

However, I do have concerns about the interpretation and contextualization of the findings. Two conceptual points deserve explicit discussion. First, the observation of trustworthy dishonesty in people with high honesty traits is arresting, as one would intuitively think that a person who scores high on honesty also acts honestly. However, this paradox is somewhat misleading. Presumably the HH trait predicts a cluster of behaviors including honesty, trustworthiness, fairness, lack of hypocrisy, lack of moral opportunism, et cetera. In the current experiment, trustworthiness appears to win out over honesty, but this could be an artifact of the task. After all, the dishonesty elicited by this task after all is very mild, and participants might even feel implicitly licensed to lie because the researchers allow them to use an external website to roll a die rather than implementing a die roll in the experiment itself. Consider a hypothetical experiment where participants are required to lie to their own mother in order to be trustworthy to a stranger: in that case one would presumably predict that higher HH yields lower trustworthy dishonesty, not higher. 'Trustworthy dishonesty' may therefore be very specific to this task and should be discussed as a special case of opposing moral motivations (see suggested literature below) rather than a novel, independent phenomenon.

Additionally, there are other motives (apart from trustworthiness) that could drive over-reporting of 3s and 4s. For example, an inequity-averse participant would prefer a 4 over a 6, but only in the high trust condition because in the no trust condition the trustor retains the \$9 remainder of the initial \$10 (if I understand correctly, please confirm). Therefore, an alternative interpretation of the findings is something like 'inequity-averse dishonesty'. Such alternative interpretations should be discussed.

These interpretation issues necessitate a better contextualization of the findings in the broader literature on cheating and conflicting motives in social decision-making. Several relevant papers that come to mind and could clarify the observations are:

- Weisel & Shalvi, PNAS, 2015. This work shows that dishonesty can emerge collaboratively, i.e. dishonesty emerges as a tool to benefit one's interaction partner. This is conceptually close to what the authors describe in the current manuscript.
- Van Baar, Chang, & Sanfey, Nature Communications, 2019. This work develops a trust game that pits inequity aversion against guilt aversion using information asymmetry, much like the opposing motives of trustworthiness and honesty in the current manuscript. It reports a pattern of moral opportunism by which participants appear to want to appear moral while maximizing payoff, a case of moral flexibility similar to trustworthy dishonesty.
- Fehr & Schmidt, Q J Econ, 1999 / Bolton & Ockenfels, AER, 2000 on inequity aversion. This motive could explain trustworthy choices in the current experiment, that is, the dishonest decision to underreport 6s may be a result of inequity aversion rather than trustworthiness. This and other alternative explanations of the observed behavior should be discussed.

Finally, the statistical method for testing overreporting (per die number) is uncommon and probably unfamiliar to many readers. It should be explained more clearly to make the manuscript more accessible to readers across fields. I am not familiar enough with the method to evaluate its use here. As an alternative, can the authors comment on whether a permutation test – simply simulating synthetic data assuming honest reporting, and then measuring whether the incidence of 6s (or any other number) in the true data exceeds the 5% most extreme simulations – might be a suitable test? This is a much more common statistical approach and might therefore make the paper more easily interpretable.

Minor points:

- How many subjects were left after data exclusions?
- Are the condition labels in table 1 flipped? The High Trust condition should be played with 30 pounds, not 3.
- The authors write “Results for other cutoffs (namely, 25% and a median split) are also reported in the Supplemental Material.” As a reviewer I did not have access to the supplement, so I cannot evaluate this statement. I think the manuscript would benefit from summarizing these findings in a brief sentence in the main text and only then referring to the supplement.
- The introduction could be streamlined. Central questions/hypotheses are introduced several times and with different phrasing/emphasis (especially concerning the existence of trustworthy dishonesty versus its link to trait HH). There are also a few spelling errors in the manuscript.

Review form: Reviewer 2

Is the manuscript scientifically sound in its present form?

Yes

Are the interpretations and conclusions justified by the results?

Yes

Is the language acceptable?

Yes

Do you have any ethical concerns with this paper?

No

Have you any concerns about statistical analyses in this paper?

Yes

Recommendation?

Major revision is needed (please make suggestions in comments)

Comments to the Author(s)

Title: Dishonesty as a sign of trustworthiness: Honesty-Humility and trustworthy dishonesty

Review:

The paper uses an innovative set-up and provides relevant insights into the linkage between personality traits and (unethical) behavior. The open disclosure of data, material and analyses is laudable. Although the paper has merit, in particular by providing some novel and interesting insights into how honesty-humility links to unethical behavior, there are several noteworthy limitations.

1. Lack of engagement with extensive literature that deals with collaborative forms of unethical behavior.

The authors repeatedly emphasize the novelty of their results, e.g. in the introduction by stating “there are no studies so far exploring whether the prosocial/ethical motivation to act trustworthily can have socially undesirable effects” and in the discussion “In the three studies we, for the first time, investigate and introduce the concept of trustworthy dishonesty and its’ personality correlates.” In the light of the large literature on collaborative forms of cheating and corruption these statements are simply not true.

For one, a growing literature has used the dyadic die rolling game (Weisel & Shalvi, 2016) that structurally closely resembles the authors’ lying-trust game in that a first mover can signal trust by reporting a high die roll and the second mover can then reciprocate the trust by matching. This literature (some references provided below) is not cited at all.

Also, studies that have examined unethical reciprocity (Leib et al., 2020) are currently not discussed. Finally and most importantly, a large literature on corruption, and in particular bribery is lacking. This literature reveals that in particular most bribery transactions crucially entail an element of trust (see for example Hunt, 2004; Jiang et al 2015; Köbis et al 2016; 2017; Shalvi et al., 2016).

Currently, the manuscript lacks a comparison and embedding of the method used and results obtained with these literatures on trust in bribery and other collaborative forms of cheating and corruption.

2. Engagement with seminal trust literature missing

Also, the manuscript would benefit from engaging with the large literature on trust and interdependence theory (e.g. Kelley et al. 2003). In the first paragraph it is important to specify what type of trust the authors refer to both when presenting the results – e.g. the study showing a link between trust and perceived corruption looks at political and interpersonal trust – and when introducing the definition, which appears to be referring to interpersonal trust.

3. Missing info about instructions of the task

From the description of the task in the results section it is not clear whether:

a) the money divided by the participants in the role of the Trustees was actually sent back to Player 1?

b) If so did the authors materialize all decisions? I assume that the authors did not, as the Trustors would receive exorbitant payoffs.

c) Most importantly, what did participants know about the materialization of their decisions? I.e. did they know how likely their decision would have financial consequences for the trustor?

This information is key and should be included in the manuscript to make it also easier for the reader to grasp what participants knew when making the decision and whether deception was used.

4. Deviations from APA guidelines

There are several deviations from APA, in particular in the results sections of Study 2 “i.e., four; $P_0 = 0.07$; 95% CI = [0.00003; 0.13]” and for Study 3 “...and fours; $P_0 = 0.06$; 95% CI = [0.002]” (emphasis with italics added). When presenting the results in accordance with APA standards, that the authors do apply in the rest of the manuscript, the results are not significant. Consistent reporting and qualification of the findings is thus advisable.

5. More info on top 10%

The authors present the results for a specific subset of the sample, namely the top 10% in Honesty-Humility. The argument for doing so is presented in the Supplementary Material. Given that the authors use this restriction to this subgroup, the manuscript should contain at least a short explanation for this choice. Moreover, in each study it should briefly be mentioned

a) how many participants fall into this category,

b) whether this subgroup differs from the rest of the sample in any identifiable way and

c) to what extend the obtained findings are robust, e.g. the authors mention that they conducted tests for the top 25% but don’t indicate in the manuscript what these analyses reveal.

To facilitate gauging the robustness of the findings without having to dig them up from the supplementary material it would be very helpful to include these bits of information in the manuscript.

6. Lack of confirmatory analyses

The authors note that “the majority of the conducted analyses were exploratory, which points toward the necessity to conduct pre-registered, confirmatory studies on the topic in the future.” That is very surprising when considering that the authors also state that “Study 2 constitutes a partial replication of Study 1 conducted in order to test the replicability of the relation between Honesty-Humility and trustworthy dishonesty in a larger sample and a different participant”

and

“Study 3 constitutes a partial replication of Study 1 and a direct replication of Study 2 on the same (in case of Study 1) and a different (in case of Study 2) panel platform, utilizing a larger sample size (in case of both studies).”

If the authors conduct two replications, how come the analyses in these studies are not confirmatory?

Minor issues:

1. The manuscript contains several typos, e.g. starting in the Abstract “We found that, when offered full trust, participants high in Honesty-Humility (the top 10%)” and several unusual formulations, e.g. the word “trustworthily” or “the trustor offered no trust”. Careful proofread and English editing is recommended.
2. Table 1 presents the payoffs for all studies using pounds. In the note it states that Study 2, used Dollars. Currently, it is not directly clear what amounts of Dollars ppts could win. Why not present the payoffs for all three studies next to each other to make it a bit clearer? This way it is also easier to grasp that the No trust condition was not included in Study 2 & 3. Also in the note you refer to “High Trust” while in the columns to “Full Trust”.
3. In the General Discussion, the sentence “Specifically, we found that, when fully trusted, participants high in Honesty-Humility chose to lie about not obtaining the outcome that would maximize their own incentive at the cost of the trustor, and/or to lie that they obtained the outcome(s) that would equalize the incentives between themselves and their trustor” Is very complicated and hard to grasp. Consider rephrasing.

Literature

- Abbink, K., Irlenbusch, B., & Renner, E. (2002). An experimental bribery game. *Journal of Law, economics, and organization*, 18(2), 428-454.
- Gross, J., Leib, M., Offerman, T., & Shalvi, S. (2018). Ethical free riding: When honest people find dishonest partners. *Psychological science*, 29(12), 1956-1968.
- Hunt, J. (2004). Trust and bribery: The role of the quid pro quo and the link with crime (No. w10510). National Bureau of Economic Research.
- Jiang, T., Lindemans, J. W., & Bicchieri, C. (2015). Can trust facilitate bribery? *Experimental evidence from China, Italy, Japan, and The Netherlands*. *Social Cognition*, 33(5), 483-504.
- Kelley, H. H., Holmes, J. G., Kerr, N. L., Reis, H. T., Rusbult, C. E., & Van Lange, P. A. (2003). *An atlas of interpersonal situations*. Cambridge University Press.
- Köbis, N. C., van Prooijen, J. W., Righetti, F., & Van Lange, P. A. (2016). Prospection in individual and interpersonal corruption dilemmas. *Review of General Psychology*, 20(1), 71-85.
- Köbis, N. C., van Prooijen, J. W., Righetti, F., & Van Lange, P. A. (2017). The road to bribery and corruption: Slippery slope or steep cliff?. *Psychological science*, 28(3), 297-306.
- Leib, M., Moran, S., & Shalvi, S. (2019). Dishonest helping and harming after (un) fair treatment. *Judgment and Decision Making*, 14(4), 423-439.
- Nichols, P. M., & Robertson, D. C. (Eds.). (2017). *Thinking about bribery: neuroscience, moral cognition and the psychology of bribery*. Cambridge University Press.
- Shalvi, S., Weisel, O., Kochavi-Gamliel, S., & Leib, M. (2016). *Corrupt Collaboration: A behavioral ethics approach*. In *Cheating, Corruption, and Concealment*
- Soraperra, I., Weisel, O., Kochavi, S., Leib, M., Shalev, H., & Shalvi, S. (2017). The bad consequences of teamwork. *Economics Letters*, 160, 12-15.
- Weisel, O., & Shalvi, S. (2015). The collaborative roots of corruption. *Proceedings of the National Academy of Sciences*, 112(34), 10651-10656.
- Wouda, J., Bijlstra, G., Frankenhuis, W. E., & Wigboldus, D. H. (2017). The collaborative roots of corruption? A replication of Weisel & Shalvi (2015).

Decision letter (RSOS-200685.R0)

Dear Ms Ścigala,

The editors assigned to your paper ("Dishonesty as a sign of trustworthiness: Honesty-Humility and trustworthy dishonesty") have now received comments from reviewers. We would like you to revise your paper in accordance with the referee and Associate Editor suggestions which can be found below (not including confidential reports to the Editor). Please note this decision does not guarantee eventual acceptance.

Please submit a copy of your revised paper before 11-Jul-2020. Please note that the revision deadline will expire at 00.00am on this date. If we do not hear from you within this time then it will be assumed that the paper has been withdrawn. In exceptional circumstances, extensions may be possible if agreed with the Editorial Office in advance. We do not allow multiple rounds of revision so we urge you to make every effort to fully address all of the comments at this stage. If deemed necessary by the Editors, your manuscript will be sent back to one or more of the original reviewers for assessment. If the original reviewers are not available, we may invite new reviewers.

- Data accessibility

If you wish to submit your supporting data or code to Dryad (<http://datadryad.org/>), or modify your current submission to dryad, please use the following link:
<http://datadryad.org/submit?journalID=RSOS&manu=RSOS-200685>

- Competing interests

- Authors' contributions

- Acknowledgements

- Funding statement

Kind regards,
Lianne Parkhouse
Editorial Coordinator
Royal Society Open Science
openscience@royalsociety.org

on behalf of Dr Inti Brazil (Associate Editor) and Essi Viding (Subject Editor)
openscience@royalsociety.org

Associate Editor's comments (Dr Inti Brazil):

I have examined the manuscript and have also received evaluations from two expert reviewers. I agree with the reviewers that the work is of interest and has potential, but that the manuscript needs to be revised substantially. There are several sections and choices that are very unclear and/or require justification, and there is also a lack of embedding within extant literature. If you believe that the concerns raised can be addressed sufficiently in a revision, please also pay close attention spelling and grammar.

Reviewers' Comments to Author:

Reviewer: 1

Comments to the Author(s)

The authors investigate a phenomenon termed 'trustworthy dishonesty', i.e. dishonesty committed in service of honoring another person's trust. They test whether the personality trait honesty-humility (HH) is predictive of such trustworthy dishonesty, which would inform the interpretation of this personality trait in a situation where trustworthiness and honesty are at odds. In study 1, they find that people who score high on HH tend to underreport payoff-maximizing events (rolling a 6) and overreport equalizing events (rolling a 3 or 4), thus exhibiting trustworthy dishonesty, more so than people with low HH. They replicate this finding in study 3 but not study 2 and note interesting differences in behavior between Prolific and MTurk samples.

In my view, the authors present an elegant experiment with interesting results. There are many papers showing that people cheat on die-rolling tasks for their own benefit, but this work shows that some people are willing to cheat even for the benefit of others. I appreciate the open science-mindedness of the authors, and I do not have major objections to the analytical approach (although I do have several questions/comments below).

However, I do have concerns about the interpretation and contextualization of the findings. Two conceptual points deserve explicit discussion. First, the observation of trustworthy dishonesty in people with high honesty traits is arresting, as one would intuitively think that a person who scores high on honesty also acts honestly. However, this paradox is somewhat misleading. Presumably the HH trait predicts a cluster of behaviors including honesty, trustworthiness, fairness, lack of hypocrisy, lack of moral opportunism, et cetera. In the current experiment, trustworthiness appears to win out over honesty, but this could be an artifact of the task. After all, the dishonesty elicited by this task after all is very mild, and participants might even feel implicitly licensed to lie because the researchers allow them to use an external website to roll a die rather than implementing a die roll in the experiment itself. Consider a hypothetical experiment where participants are required to lie to their own mother in order to be trustworthy to a stranger: in that case one would presumably predict that higher HH yields lower trustworthy dishonesty, not higher. 'Trustworthy dishonesty' may therefore be very specific to this task and should be discussed as a special case of opposing moral motivations (see suggested literature below) rather than a novel, independent phenomenon.

Additionally, there are other motives (apart from trustworthiness) that could drive over-reporting of 3s and 4s. For example, an inequity-averse participant would prefer a 4 over a 6, but only in the high trust condition because in the no trust condition the trustor retains the \$9 remainder of the initial \$10 (if I understand correctly, please confirm). Therefore, an alternative interpretation of the findings is something like 'inequity-averse dishonesty'. Such alternative interpretations should be discussed.

These interpretation issues necessitate a better contextualization of the findings in the broader literature on cheating and conflicting motives in social decision-making. Several relevant papers that come to mind and could clarify the observations are:

- Weisel & Shalvi, PNAS, 2015. This work shows that dishonesty can emerge collaboratively, i.e. dishonesty emerges as a tool to benefit one's interaction partner. This is conceptually close to what the authors describe in the current manuscript.
- Van Baar, Chang, & Sanfey, Nature Communications, 2019. This work develops a trust game that pits inequity aversion against guilt aversion using information asymmetry, much like the opposing motives of trustworthiness and honesty in the current manuscript. It reports a pattern of moral opportunism by which participants appear to want to appear moral while maximizing payoff, a case of moral flexibility similar to trustworthy dishonesty.
- Fehr & Schmidt, Q J Econ, 1999 / Bolton & Ockenfels, AER, 2000 on inequity aversion. This motive could explain trustworthy choices in the current experiment, that is, the dishonest

decision to underreport 6s may be a result of inequity aversion rather than trustworthiness. This and other alternative explanations of the observed behavior should be discussed.

Finally, the statistical method for testing overreporting (per die number) is uncommon and probably unfamiliar to many readers. It should be explained more clearly to make the manuscript more accessible to readers across fields. I am not familiar enough with the method to evaluate its use here. As an alternative, can the authors comment on whether a permutation test – simply simulating synthetic data assuming honest reporting, and then measuring whether the incidence of 6s (or any other number) in the true data exceeds the 5% most extreme simulations – might be a suitable test? This is a much more common statistical approach and might therefore make the paper more easily interpretable.

Minor points:

- How many subjects were left after data exclusions?
- Are the condition labels in table 1 flipped? The High Trust condition should be played with 30 pounds, not 3.
- The authors write “Results for other cutoffs (namely, 25% and a median split) are also reported in the Supplemental Material.” As a reviewer I did not have access to the supplement, so I cannot evaluate this statement. I think the manuscript would benefit from summarizing these findings in a brief sentence in the main text and only then referring to the supplement.
- The introduction could be streamlined. Central questions/hypotheses are introduced several times and with different phrasing/emphasis (especially concerning the existence of trustworthy dishonesty versus its link to trait HH). There are also a few spelling errors in the manuscript.

Reviewer: 2

Comments to the Author(s)

Title: Dishonesty as a sign of trustworthiness: Honesty-Humility and trustworthy dishonesty

Review:

The paper uses an innovative set-up and provides relevant insights into the linkage between personality traits and (unethical) behavior. The open disclosure of data, material and analyses is laudable. Although the paper has merit, in particular by providing some novel and interesting insights into how honesty-humility links to unethical behavior, there are several noteworthy limitations.

1. Lack of engagement with extensive literature that deals with collaborative forms of unethical behavior.

The authors repeatedly emphasize the novelty of their results, e.g. in the introduction by stating “there are no studies so far exploring whether the prosocial/ethical motivation to act trustworthily can have socially undesirable effects” and in the discussion “In the three studies we, for the first time, investigate and introduce the concept of trustworthy dishonesty and its’ personality correlates.” In the light of the large literature on collaborative forms of cheating and corruption these statements are simply not true.

For one, a growing literature has used the dyadic die rolling game (Weisel & Shalvi, 2016) that structurally closely resembles the authors’ lying-trust game in that a first mover can signal trust by reporting a high die roll and the second mover can then reciprocate the trust by matching. This literature (some references provided below) is not cited at all.

Also, studies that have examined unethical reciprocity (Leib et al., 2020) are currently not discussed. Finally and most importantly, a large literature on corruption, and in particular bribery is lacking. This literature reveals that in particular most bribery transactions crucially entail an element of trust (see for example Hunt, 2004; Jiang et al 2015; Köbis et al 2016; 2017; Shalvi et al., 2016).

Currently, the manuscript lacks a comparison and embedding of the method used and results obtained with these literatures on trust in bribery and other collaborative forms of cheating and corruption.

2. Engagement with seminal trust literature missing

Also, the manuscript would benefit from engaging with the large literature on trust and interdependence theory (e.g. Kelley et al. 2003). In the first paragraph it is important to specify what type of trust the authors refer to both when presenting the results – e.g. the study showing a link between trust and perceived corruption looks at political and interpersonal trust – and when introducing the definition, which appears to be referring to interpersonal trust.

3. Missing info about instructions of the task

From the description of the task in the results section it is not clear whether:

a) the money divided by the participants in the role of the Trustees was actually sent back to Player 1?

b) If so did the authors materialize all decisions? I assume that the authors did not, as the Trustors would receive exorbitant payoffs.

c) Most importantly, what did participants know about the materialization of their decisions? I.e. did they know how likely their decision would have financial consequences for the trustor?

This information is key and should be included in the manuscript to make it also easier for the reader to grasp what participants knew when making the decision and whether deception was used.

4. Deviations from APA guidelines

There are several deviations from APA, in particular in the results sections of Study 2 “i.e., four; $P_o = 0.07$; 95% CI = [0.00003; 0.13]” and for Study 3 “...and fours; $P_o = 0.06$; 95% CI = [0.002]” (emphasis with italics added). When presenting the results in accordance with APA standards, that the authors do apply in the rest of the manuscript, the results are not significant. Consistent reporting and qualification of the findings is thus advisable.

5. More info on top 10%

The authors present the results for a specific subset of the sample, namely the top 10% in Honesty-Humility. The argument for doing so is presented in the Supplementary Material. Given that the authors use this restriction to this subgroup, the manuscript should contain at least a short explanation for this choice. Moreover, in each study it should briefly be mentioned

a) how many participants fall into this category,

b) whether this subgroup differs from the rest of the sample in any identifiable way and

c) to what extent the obtained findings are robust, e.g. the authors mention that they conducted tests for the top 25% but don't indicate in the manuscript what these analyses reveal.

To facilitate gauging the robustness of the findings without having to dig them up from the supplementary material it would be very helpful to include these bits of information in the manuscript.

6. Lack of confirmatory analyses

The authors note that “the majority of the conducted analyses were exploratory, which points toward the necessity to conduct pre-registered, confirmatory studies on the topic in the future.”

That is very surprising when considering that the authors also state that

“Study 2 constitutes a partial replication of Study 1 conducted in order to test the replicability of the relation between Honesty-Humility and trustworthy dishonesty in a larger sample and a different participant”

and

“Study 3 constitutes a partial replication of Study 1 and a direct replication of Study 2 on the same (in case of Study 1) and a different (in case of Study 2) panel platform, utilizing a larger sample size (in case of both studies).”

If the authors conduct two replications, how come the analyses in these studies are not confirmatory?

Minor issues:

1. The manuscript contains several typos, e.g. starting in the Abstract “We found that, when offered full trust, participants high in Honesty-Humility (the top 10%)” and several unusual

formulations, e.g. the word “trustworthily” or “the trustor offered no trust”. Careful proofread and English editing is recommended.

2. Table 1 presents the payoffs for all studies using pounds. In the note it states that Study 2, used Dollars. Currently, it is not directly clear what amounts of Dollars ppts could win. Why not present the payoffs for all three studies next to each other to make it a bit clearer? This way it is also easier to grasp that the No trust condition was not included in Study 2 & 3. Also in the note you refer to ““High Trust” while in the columns to “Full Trust”.

3. In the General Discussion, the sentence “Specifically, we found that, when fully trusted, participants high in Honesty-Humility chose to lie about not obtaining the outcome that would maximize their own incentive at the cost of the trustor, and/or to lie that they obtained the outcome(s) that would equalize the incentives between themselves and their trustor” Is very complicated and hard to grasp. Consider rephrasing.

Literature

- Abbink, K., Irlenbusch, B., & Renner, E. (2002). An experimental bribery game. *Journal of Law, economics, and organization*, 18(2), 428-454.
- Gross, J., Leib, M., Offerman, T., & Shalvi, S. (2018). Ethical free riding: When honest people find dishonest partners. *Psychological science*, 29(12), 1956-1968.
- Hunt, J. (2004). Trust and bribery: The role of the quid pro quo and the link with crime (No. w10510). National Bureau of Economic Research.
- Jiang, T., Lindemans, J. W., & Bicchieri, C. (2015). Can trust facilitate bribery? Experimental evidence from China, Italy, Japan, and The Netherlands. *Social Cognition*, 33(5), 483-504.
- Kelley, H. H., Holmes, J. G., Kerr, N. L., Reis, H. T., Rusbult, C. E., & Van Lange, P. A. (2003). *An atlas of interpersonal situations*. Cambridge University Press.
- Köbis, N. C., van Prooijen, J. W., Righetti, F., & Van Lange, P. A. (2016). Prospection in individual and interpersonal corruption dilemmas. *Review of General Psychology*, 20(1), 71-85.
- Köbis, N. C., van Prooijen, J. W., Righetti, F., & Van Lange, P. A. (2017). The road to bribery and corruption: Slippery slope or steep cliff?. *Psychological science*, 28(3), 297-306.
- Leib, M., Moran, S., & Shalvi, S. (2019). Dishonest helping and harming after (un) fair treatment. *Judgment and Decision Making*, 14(4), 423-439.
- Nichols, P. M., & Robertson, D. C. (Eds.). (2017). *Thinking about bribery: neuroscience, moral cognition and the psychology of bribery*. Cambridge University Press.
- Shalvi, S., Weisel, O., Kochavi-Gamliel, S., & Leib, M. (2016). Corrupt Collaboration: A behavioral ethics approach. In *Cheating, Corruption, and Concealment*
- Soraperra, I., Weisel, O., Kochavi, S., Leib, M., Shalev, H., & Shalvi, S. (2017). The bad consequences of teamwork. *Economics Letters*, 160, 12-15.
- Weisel, O., & Shalvi, S. (2015). The collaborative roots of corruption. *Proceedings of the National Academy of Sciences*, 112(34), 10651-10656.
- Wouda, J., Bijlstra, G., Frankenhuys, W. E., & Wigboldus, D. H. (2017). The collaborative roots of corruption? A replication of Weisel & Shalvi (2015).

Author's Response to Decision Letter for (RSOS-200685.R0)

See Appendix A.

RSOS-200685.R1 (Revision)

Review form: Reviewer 1

Is the manuscript scientifically sound in its present form?

Yes

Are the interpretations and conclusions justified by the results?

Yes

Is the language acceptable?

Yes

Do you have any ethical concerns with this paper?

No

Have you any concerns about statistical analyses in this paper?

No

Recommendation?

Accept as is

Comments to the Author(s)

My concerns have been addressed and I support publication in the current form.

Review form: Reviewer 2

Is the manuscript scientifically sound in its present form?

Yes

Are the interpretations and conclusions justified by the results?

Yes

Is the language acceptable?

Yes

Do you have any ethical concerns with this paper?

No

Have you any concerns about statistical analyses in this paper?

No

Recommendation?

Accept as is

Comments to the Author(s)

The authors address all raised issues.

It is a very nice manuscript

Decision letter (RSOS-200685.R1)

Dear Ms Ścigala,

It is a pleasure to accept your manuscript entitled "Dishonesty as a signal of trustworthiness: Honesty-Humility and trustworthy dishonesty" in its current form for publication in Royal Society Open Science. The comments of the reviewer(s) who reviewed your manuscript are included at the foot of this letter.

Please ensure that you send to the editorial office an editable version of your accepted manuscript, and individual files for each figure and table included in your manuscript. You can send these in a zip folder if more convenient. Failure to provide these files may delay the processing of your proof.

on behalf of Dr Inti Brazil (Associate Editor) and Essi Viding (Subject Editor)
openscience@royalsociety.org

Reviewer comments to Author:
Reviewer: 1

Comments to the Author(s)
My concerns have been addressed and I support publication in the current form.

Reviewer: 2

Comments to the Author(s)
The authors address all raised issues.
It is a very nice manuscript

Appendix A

Associate Editor's comments:

I have examined the manuscript and have also received evaluations from two expert reviewers. I agree with the reviewers that the work is of interest and has potential, but that the manuscript needs to be revised substantially. There are several sections and choices that are very unclear and/or require justification, and there is also a lack of embedding within extant literature. If you believe that the concerns raised can be addressed sufficiently in a revision, please also pay close attention spelling and grammar.

Response:

Dear Professor Brazil,

Thank you very much for the evaluation of our manuscript and the invitation to submit a revised version. Also, we are very grateful for the constructive comments provided by you and the reviewers, which have, as we believe, helped us in further strengthening the manuscript. We have thoroughly revised the manuscript based on the feedback. We have elaborated on the aspects of the manuscript that required further justification and we extended the embedding of our investigation with the existing literature. Further, we carefully double-checked for our spelling and grammar. In the following, we will provide point-by-point responses to each of the comments.

In addition, following the submission guidelines, we provide a tracked-changes version of the Manuscript in the end of this document. Please note that the number of participants and the statistical indices are slightly modified. This is because we noticed that a few participants who should have been excluded from the study, were still in the datasets. We are sorry for this mistake. Also, please note that these modifications did not change the results on a conceptual level.

Sincerely,
Karolina Ścigała, Christoph Schild, and Ingo Zettler

Reviewers' Comments to Author:

Reviewer: 1

Comments to the Author(s)

The authors investigate a phenomenon termed 'trustworthy dishonesty', i.e. dishonesty committed in service of honoring another person's trust. They test whether the personality trait honesty-humility (HH) is predictive of such trustworthy dishonesty, which would inform the interpretation of this personality trait in a situation where trustworthiness and honesty are at odds. In study 1, they find that people who score high on HH tend to underreport

payoff-maximizing events (rolling a 6) and overreport equalizing events (rolling a 3 or 4), thus exhibiting trustworthy dishonesty, more so than people with low HH. They replicate this finding in study 3 but not study 2 and note interesting differences in behavior between Prolific and MTurk samples.

In my view, the authors present an elegant experiment with interesting results. There are many papers showing that people cheat on die-rolling tasks for their own benefit, but this work shows that some people are willing to cheat even for the benefit of others. I appreciate the open science-mindedness of the authors, and I do not have major objections to the analytical approach (although I do have several questions/comments below).

1. However, I do have concerns about the interpretation and contextualization of the findings. Two conceptual points deserve explicit discussion. First, the observation of trustworthy dishonesty in people with high honesty traits is arresting, as one would intuitively think that a person who scores high on honesty also acts honestly. However, this paradox is somewhat misleading. Presumably the HH trait predicts a cluster of behaviors including honesty, trustworthiness, fairness, lack of hypocrisy, lack of moral opportunism, et cetera. In the current experiment, trustworthiness appears to win out over honesty, but this could be an artifact of the task. After all, the dishonesty elicited by this task after all is very mild, and participants might even feel implicitly licensed to lie because the researchers allow them to use an external website to roll a die rather than implementing a die roll in the experiment itself. Consider a hypothetical experiment where participants are required to lie to their own mother in order to be trustworthy to a stranger: in that case one would presumably predict that higher HH yields lower trustworthy dishonesty, not higher. 'Trustworthy dishonesty' may therefore be very specific to this task and should be discussed as a special case of opposing moral motivations (see suggested literature below) rather than a novel, independent phenomenon.

Response:

First of all, thank you very much for your positive evaluation overall and your thoughtful comments.

We fully agree with your theorizing that the observed link between HH and trustworthy dishonesty might be tied to specific situations, and, in turn, the specific design and measures of a study. Correspondingly, we now discuss the possibility that trustworthy dishonesty might be an artifact of the task in the "Discussion" section:

"Third, our results might be attributed to the characteristics of the task. Specifically, participants high in Honesty-Humility might have chosen to engage in trustworthy dishonesty (in our Studies 1 and 3) because being asked to roll a die on an anonymous, external website might have been interpreted as an implicit license to lie (e.g., Lilleholt et al., 2020) and hence as

ethically acceptable in the context of the task.” (p. 28)

Further, it is fully correct that Honesty-Humility (irrespective of its name) does not comprise honesty-related trait characteristics only, but rather, more broadly, a mélange of anti- or pro-social trait characteristics including cooperativeness, fairness, modesty, and sincerity (indeed, some researchers have thus argued that the term Honesty-Humility is not the most accurate one for this basic trait, see, e.g., Diebels et al., 2018). We now briefly point at this in the following:

“One of the basic personality traits that has been consistently positively related to a broad array of prosocial behaviors (including both trustworthiness and honesty, but also cooperativeness, fairness, and modesty; for meta-analytical findings, see Thielmann et al., 2020; Zettler et al., 2020) is Honesty-Humility from the HEXACO Model of Personality. Honesty-Humility is defined as “the tendency to be fair and genuine in dealing with others, in the sense of cooperating with others even when one might exploit them without suffering retaliation” (Ashton & Lee, 2007; p. 156).” (pp. 6-7)

In addition, following your suggestion as well as in responding to your comment #3 (see below), we discuss our findings also in the light of other findings on opposing moral motivations.

2. Additionally, there are other motives (apart from trustworthiness) that could drive over-reporting of 3s and 4s. For example, an inequity-averse participant would prefer a 4 over a 6, but only in the high trust condition because in the no trust condition the trustor retains the \$9 remainder of the initial \$10 (if I understand correctly, please confirm). Therefore, an alternative interpretation of the findings is something like ‘inequity-averse dishonesty’. Such alternative interpretations should be discussed.

Response:

Indeed, the trustors retained the \$/€9 from the initial \$/€10 in the No Trust condition, and they retained \$/€0 from the initial \$/€10 in the Full Trust condition. Again, we agree that your alternative explanation might also explain our findings, and thus now discuss the potential role of inequity-aversion in the “Discussion” section:

“First of all, it should be noted that inequity-aversion (e.g., Bolton & Ockenfels, 2000; van Baar et al., 2019) might have played a role in our findings since it has already been found to play a role in trust games (van Baar et al., 2019). Specifically, participants who engaged in other-benefitting dishonesty in the Full Trust condition increased equity between themselves and their partner (because their trustors did not keep anything from their initial endowment), whereas participants who engaged in other-benefitting dishonesty in the No Trust condition decreased equity between themselves and their partner (because their trustors kept 90% of their initial endowment). Consequently, it might be that it was inequity-aversion rather than

trustworthiness (or a mixture of both) that motivated participants to engage in other-benefitting dishonesty to a larger degree in the Full Trust rather than in the No Trust condition.” (pp. 27-28)

- 3. These interpretation issues necessitate a better contextualization of the findings in the broader literature on cheating and conflicting motives in social decision-making. Several relevant papers that come to mind and could clarify the observations are:**
- Weisel & Shalvi, PNAS, 2015. This work shows that dishonesty can emerge collaboratively, i.e. dishonesty emerges as a tool to benefit one’s interaction partner. This is conceptually close to what the authors describe in the current manuscript.**
 - Van Baar, Chang, & Sanfey, Nature Communications, 2019. This work develops a trust game that pits inequity aversion against guilt aversion using information asymmetry, much like the opposing motives of trustworthiness and honesty in the current manuscript. It reports a pattern of moral opportunism by which participants appear to want to appear moral while maximizing payoff, a case of moral flexibility similar to trustworthy dishonesty.**
 - Fehr & Schmidt, Q J Econ, 1999 / Bolton & Ockenfels, AER, 2000 on inequity aversion. This motive could explain trustworthy choices in the current experiment, that is, the dishonest decision to underreport 6s may be a result of inequity aversion rather than trustworthiness. This and other alternative explanations of the observed behavior should be discussed.**

Response:

Thank you very much for these recommendations. We now include/discuss (although sometimes briefly only) further work on corruption, bribery, and trust that might be relevant for (the interpretation of) our investigation

“Despite the vast array of research pointing towards social desirability of trustworthiness, studies on corruption and bribery suggest that the motivation to act in a trustworthy way might also lead to socially undesirable behavior including dishonesty (Abbink, 2002; Gross et al., 2018; Hunt, 2004; Jiang et al., 2015; Köbis et al., 2016, 2017; Ścigala et al., 2019; Soraperra et al., 2017, 2017; Weisel & Shalvi, 2015). Indeed, forming and maintaining successful corrupt interactions obviously requires that the involved parties behave in a trustworthy way with regard to each other (e.g., reciprocate the corrupt offer and refrain from reporting the illegal transaction(s) to authorities; Dungan et al., 2014; Köbis et al., 2016; Lamsdorff & Frank, 2011). For instance, Jiang et al. (2015) found that high trust was related to more bribery in otherwise low-corruption countries. In another study, Abbink (2004) used a multi-round bribery game, in which participants were either matched with a different interaction partner every round of the game (the “strangers” condition) or had a fixed partner throughout the game (the “partners” condition). Participants in the “strangers” condition engaged in less bribery than participants in

the “partners” condition, arguably because they might not have been able to develop as much trust in their interaction partner in the former condition as compared to the latter. Similarly, Weisel and Shalvi (2015; replicated by Wouda et al., 2017) found that when participants had to collaboratively engage in dishonesty to increase both their own and their partner’s profits, the cheating rates were substantially higher as compared to when they could individually engage in dishonesty to increase only their own or only their partner’s profits. Arguably, the increased rates of dishonesty in the collaborative context, relative to the individual context, might have been caused by a motivation to behave in a trustworthy way towards one’s interaction partner. In sum, the motivation to act in a trustworthy way might lead to socially undesirable behavior (i.e., dishonesty) when expressed in the context of corruption and bribery.” (pp. 5-6)

In addition, we compare trustworthy dishonesty to corruption and bribery in the following:

“Herein, we will examine if participants high in Honesty-Humility are willing to lie in order to fulfill their trustors’ expectations regarding their trustworthiness (i.e., engage in *trustworthy dishonesty*). In doing so, we explore if people high in Honesty-Humility engage in other-benefitting dishonesty as a mean to act trustworthily when benefiting the other is at odds with their self-interest. Please note that we focus on the type of dishonesty in which benefiting the other is at odds with benefiting oneself in order to assure that the motivation to act trustworthily is not confounded with one’s self-interest (as it is the case in bribery and corruption settings in which acting trustworthily towards one’s interaction partner and self-interests are typically aligned with each other; e.g. Abbink, 2004; Jiang et al., 2015; Weisel & Shalvi, 2015). In other words, we focus on a situation with conflicting interests between the trustor and the trustee because aligning interests between them might not give the trustee an opportunity to signal their trustworthiness without signaling their self-interest as well (Balliet & van Lange, 2013; van Lange & Balliet, 2015).” (p. 7)

Furthermore, we discuss the possibility that our findings might be attributed to inequity aversion (e.g., Bolton & Ockenfels, 2000; Van Baar et al., 2019) in the “Discussion” section (see our response to your comment #1).

4. Finally, the statistical method for testing overreporting (per die number) is uncommon and probably unfamiliar to many readers. It should be explained more clearly to make the manuscript more accessible to readers across fields. I am not familiar enough with the method to evaluate its use here. As an alternative, can the authors comment on whether a permutation test—simply simulating synthetic data assuming honest reporting, and then measuring whether the incidence of 6s (or any other number) in the true data exceeds the 5% most extreme simulations—might be a suitable test? This is a much more common statistical approach and might therefore make the paper more easily interpretable.

Response

Response:

Thank you for your suggestion. We decided not to perform a permutation test and we provide further details on why the method for testing over- and under- reporting is suitable.

“Because the method used above (i.e., the plots) does not allow to calculate the proportion of underreporting/overreporting dishonest individuals, and is purely based on figure interpretation, we employed another analytical procedure following an approach suggested by Moshagen and Hilbig (2017). This procedure lets us estimate the proportions of participants who engaged in dishonest under- and over-reporting. Using this procedure is necessary because the die-roll outcomes obtained in the lying-trust game are anonymous and hence we cannot determine which participants actually rolled the numbers they reported and which participants did not (this kind of anonymity is necessary to give participants adequate conditions for cheating; Schild et al., 2019). In other words, the number of participants who indicated that they rolled a given number reflects both the number of participants who actually rolled the number and the number of participants who dishonestly misreported the number. Hence, to obtain an accurate proportion of dishonest individuals, we have to take into account the objective probability of rolling each number in one die roll (which, in our study, equals 1/6; Moshagen and Hilbig, 2017). Please note that this method has been shown to outperform traditional analytical approaches (Moshagen and Hilbig, 2017) and is increasingly used in studies on dishonesty (Heck et al., 2018; Köbis et al., 2019; Moshagen et al., 2018).” (pp. 14-15)

Minor points:

5. How many subjects were left after data exclusions?

Response:

The numbers of participants provided at the beginning of each “Procedure and participants” sections are numbers after exclusions (based on the attention checks and the control questions). We now also include the numbers of participants who were excluded based on the attention checks and the control questions.

Study 1:

“Participants who failed the attention checks interspersed within the questionnaires ($N = 17$) or failed the control questions before the lying-trust game ($N = 87$) were not included in the analyses.” (pp. 8-9)

Study 2:

“Participants who failed the attention checks interspersed within the questionnaires ($N = 73$) or failed the control questions before the lying-trust game ($N = 135$) were not included in the analyses.” (p. 20)

Study 3:

“Participants who failed the attention checks interspersed within the questionnaires ($N = 21$) or failed the control questions before the lying-trust game ($N = 130$) were not included in the analyses.” (p. 24)

6. Are the condition labels in table 1 flipped? The High Trust condition should be played with 30 pounds, not 3.

Response:

Indeed, the condition labels were mistakenly flipped. Thank you very much for spotting this. We have now corrected the table (pp. 11-12).

7. The authors write “Results for other cutoffs (namely, 25% and a median split) are also reported in the Supplemental Material.” As a reviewer I did not have access to the supplement, so I cannot evaluate this statement. I think the manuscript would benefit from summarizing these findings in a brief sentence in the main text and only then referring to the supplement.

Response:

We thought that the Supplemental Material was available for the reviewers and we apologize if that was not the case. We make the Supplemental Material available in the OSF (https://osf.io/ywgp5/?view_only=8a0e9577b87f4c77a6faf40331f5b6c1). In addition, we now provide a short summary of the results for the other cutoffs in the Results section of each study:

Study 1:

“Please note that the detailed results for other cutoffs (namely, 25% and a median split) are reported in the Supplemental Material. In brief, similarly to participants in the top 10% Honesty-Humility, both participants in the top 25% Honesty-Humility and with Honesty-Humility values higher than the median underreported the income-maximizing outcome (six) in the Full Trust condition, but not in the No Trust condition, and overreported the equalizing outcome (four) in the Full Trust condition, but not in the No Trust condition.” (p. 17)

Study 2:

“Please note that the detailed results for other cutoffs (namely, 25% and a median split) are reported in the Supplemental Material. In brief, participants high in Honesty-Humility (both in the top 25% and with values higher than the median) overreported the equalizing outcome (four) and underreported the income-minimizing outcomes (ones and twos) similarly to participants in the top 10%.” (p. 21)

Study 3:

“Please note that the detailed results for other cutoffs (namely, 25% and a median split) are reported in the Supplemental Material. In brief, similarly to participants in the top 10%, both participants in the top 25% and with values higher than the median overreported the equalizing outcomes (threes and fours). On the other hand, neither participants in the top 25%, nor with values higher than the median underreported the income-maximizing outcome (six).” (p. 25)

8. The introduction could be streamlined. Central questions/hypotheses are introduced several times and with different phrasing/emphasis (especially concerning the existence of trustworthy dishonesty versus its link to trait HH). There are also a few spelling errors in the manuscript.

Response:

We have now streamlined the introduction (pp. 3-8) and paid particular attention to avoid spelling mistakes.

Reviewer: 2

Comments to the Author(s)

Title: Dishonesty as a sign of trustworthiness: Honesty-Humility and trustworthy dishonesty

Review:

The paper uses an innovative set-up and provides relevant insights into the linkage between personality traits and (unethical) behavior. The open disclosure of data, material and analyses is laudable. Although the paper has merit, in particular by providing some novel and interesting insights into how honesty-humility links to unethical behavior, there are several noteworthy limitations.

1. Lack of engagement with extensive literature that deals with collaborative forms of unethical behavior.

The authors repeatedly emphasize the novelty of their results, e.g. in the introduction by stating “there are no studies so far exploring whether the prosocial/ethical motivation to act trustworthily can have socially undesirable effects” and in the discussion “In the three

studies we, for the first time, investigate and introduce the concept of trustworthy dishonesty and its' personality correlates.” In the light of the large literature on collaborative forms of cheating and corruption these statements are simply not true. For one, a growing literature has used the dyadic die rolling game (Weisel & Shalvi, 2016) that structurally closely resembles the authors' lying-trust game in that a first mover can signal trust by reporting a high die roll and the second mover can then reciprocate the trust by matching. This literature (some references provided below) is not cited at all. Also, studies that have examined unethical reciprocity (Leib et al., 2020) are currently not discussed. Finally and most importantly, a large literature on corruption, and in particular bribery is lacking. This literature reveals that in particular most bribery transactions crucially entail an element of trust (see for example Hunt, 2004; Jiang et al 2015; Köbis et al 2016; 2017; Shalvi et al., 2016). Currently, the manuscript lacks a comparison and embedding of the method used and results obtained with these literatures on trust in bribery and other collaborative forms of cheating and corruption.

Response:

Thank you very much for these suggestion. As also described in our response to comment #3 by Reviewer 1, we now describe relevant literature on corruption and bribery and link (the interpretation of) our investigation to this literature (see, response to comment #3 by Reviewer 1).

In addition to what is described in our response above, we refer to findings by Leib et al. (2019) in the “Discussion” section:

“Second, Leib et al. (2019) found that people engaged in dishonest helping to a larger degree after a fair as compared to an unfair treatment. In line with this, it is possible that participants in the No Trust condition engaged in less other-benefitting dishonesty as compared to participants in the Full Trust condition because participants in the No Trust condition perceived the amount of money they received from the trustee as unfair.” (pp. 27-28)

2. Engagement with seminal trust literature missing

Also, the manuscript would benefit from engaging with the large literature on trust and interdependence theory (e.g. Kelley et al. 2003). In the first paragraph it is important to specify what type of trust the authors refer to both when presenting the results – e.g. the study showing a link between trust and perceived corruption looks at political and interpersonal trust – and when introducing the definition, which appears to be referring to interpersonal trust.

Response:

Again, thank you very much for these hints. We now clarify that we refer to interpersonal trust and we include literature on trust and interdependence theory

“Interpersonal trust is fundamental for the effective functioning of social interactions as well as of society as a whole. It has been found to be related to many societal outcomes such as lower corruption perception (e.g., Morris & Klesner, 2010), higher economic growth (Knack & Keefer, 1997), and more efficient judicial systems (Porta et al., 1996). Among the many definitions of interpersonal trust, one of the most comprehensive ones defines it as “a risky choice of making oneself dependent on the actions of another in a situation of uncertainty, based upon some expectation of whether the other will act in a benevolent fashion despite an opportunity to betray” (Thielmann & Hilbig, 2015a; p. 251). Thus, (interpersonal) trust is based on an interaction that includes a trustor (the one that decides to trust) and a trustee (the one that may or may not behave trustworthily). Because of this interdependence, trusting can be beneficial for trustors only when trustees fulfill the trustors’ expectations regarding their behavior (i.e., behave trustworthily; Kelley et al., 2003; Levine et al., 2018; Thielmann & Hilbig, 2015b), which can be influenced by many factors such as behavioral control, power imbalance, and conflict of interest (e.g., Kelley et al., 2003; van Lange & Balliet, 2015). For instance, one’s trustworthiness could be expressed in returning a loan borrowed from a friend, remaining faithful to one’s partner, or keeping a promise to take good care of one’s neighbor’s cat when the neighbor is away.” (p. 3)

3. Missing info about instructions of the task

From the description of the task in the results section it is not clear whether:

- a) the money divided by the participants in the role of the Trustees was actually sent back to Player 1?**
- b) If so did the authors materialize all decisions? I assume that the authors did not, as the Trustors would receive exorbitant payoffs.**
- c) Most importantly, what did participants know about the materialization of their decisions? I.e. did they know how likely their decision would have financial consequences for the trustor?**

This information is key and should be included in the manuscript to make it also easier for the reader to grasp what participants knew when making the decision and whether deception was used.

Response:

We now provide the missing information about the task:

“The bonuses based on the decisions in the lying-trust game were materialized for 50 dyads involved in each study. In all studies, participants knew before they made their decisions that

their decisions have consequences for themselves as well as for the participants they were matched with and that the bonuses are based on the above mentioned lottery.” (p. 11)

4. Deviations from APA guidelines

There are several deviations from APA, in particular in the results sections of Study 2 “i.e., four; $P_o = 0.07$; 95% CI = [0.00003; 0.13]” and for Study 3 “...and fours; $P_o = 0.06$; 95% CI = [0.002]” (emphasis with italics added). When presenting the results in accordance with APA standards, that the authors do apply in the rest of the manuscript, the results are not significant. Consistent reporting and qualification of the findings is thus advisable.

Response:

We realize that we report more places after decimal than we do in other cases; however we are not aware of another way to report such findings, because the numbers indicating the lower bound of the confidence interval are higher than zero. Thus, we cannot report that the CIs include 0, and, in turn, that the results should be considered as non-significant. However, we added to the Discussion section that the lower bound of these CIs are just above 0.

“Finally, the lower bounds of the confidence intervals for several of our significant findings were only slightly higher than zero (which would indicate an insignificant finding). This points towards the necessity of future replication of our findings.” (p. 29)

5. More info on top 10%

The authors present the results for a specific subset of the sample, namely the top 10% in Honesty-Humility. The argument for doing so is presented in the Supplementary Material. Given that the authors use this restriction to this subgroup, the manuscript should contain at least a short explanation for this choice. Moreover, in each study it should briefly be mentioned

a) how many participants fall into this category,

b) whether this subgroup differs from the rest of the sample in any identifiable way and

c) to what extent the obtained findings are robust, e.g. the authors mention that they conducted tests for the top 25% but don't indicate in the manuscript what these analyses reveal.

To facilitate gauging the robustness of the findings without having to dig them up from the supplementary material it would be very helpful to include these bits of information in the manuscript.

Response:

We now provide a brief explanation about why we chose the 10% threshold:

“The detailed reasoning behind the choice of the 10% cutoff is outlined in the Supplemental Material (pp. 1-5). In brief, based on visual examination of the data, we concluded that—in Studies 1 and 3—participants with the top 10% scores in Honesty-Humility consistently underreported the outcome that maximized their own income and minimized their partner’s income.”

In addition, we provide the additional information about the cutoffs:

- a) We indicate the number of participants who fall in this category (the 10% category) in the text (p. 15; p. 21; p. 24) as well as in the descriptions of Tables 2-4.
- b) We now specify how the top 10% differs from the rest of the sample:
“Finally, participants in the top 10% in Honesty-Humility were older and included more women than the remaining 90% (for details, see Supplemental Material, p. 20), which might have influenced the results (based on meta-analytic results, both older participants and women generally cheat less than younger participants and men, respectively; Gerlach et al., 2019).” (p. 28)
- c) We now summarize the results for other thresholds in the “Results” sections of the Manuscript (see, response to comment #7 by Reviewer 1).

6. Lack of confirmatory analyses

The authors note that “the majority of the conducted analyses were exploratory, which points toward the necessity to conduct pre-registered, confirmatory studies on the topic in the future.” That is very surprising when considering that the authors also state that “Study 2 constitutes a partial replication of Study 1 conducted in order to test the replicability of the relation between Honesty-Humility and trustworthy dishonesty in a larger sample and a different participant”

and

“Study 3 constitutes a partial replication of Study 1 and a direct replication of Study 2 on the same (in case of Study 1) and a different (in case of Study 2) panel platform, utilizing a larger sample size (in case of both studies).”

If the authors conduct two replications, how come the analyses in these studies are not confirmatory?

Response:

We described these issues in the Supplemental Material. We include the explanation behind deviating from the confirmatory analyses below:

“Deviations from the preregistrations

Study 1: the initial aims

The initial aim of Study 1 was to test how Honesty-Humility (Ashton & Lee, 2008) and Guilt Proneness (Tangney & Dearing, 2004) differ with regard to cheating for the sake of the trustor when offered full trust (preregistered as “High Vulnerability”) as compared to no trust (preregistered as “Low Vulnerability”; see, https://osf.io/s2k59/?view_only=7ba46a5f2bdf45a7bc763011c1c346f6; masked for review). The idea to explore the phenomena of trustworthy dishonesty arose after data collection when we observed that some participants engaged in trustworthy dishonesty as a response to being offered full trust by their trustor (see, Tables 1a-2b).

Studies 1-3: the reasoning behind the 10% threshold

In the preregistrations for both Studies 2 and 3 (see https://osf.io/dtk93/?view_only=907fccbcc7b04bf694caa061ba83199e and https://osf.io/x8mz3/?view_only=7dc346f22ac4451892deec17035a2fcb, respectively; masked for review), we planned to compute the proportion of individuals dishonestly underreporting the income-maximizing outcome for the entire sample (Hypotheses 1 in both studies) and for the participants above and below the median of Honesty-Humility (Hypotheses 2 in both studies). These hypotheses were based on Study 1 where we observed a significant proportion of such individuals in the Full Trust condition ($P = 0.21$; 95% CI [0.07; 0.35]). In addition, we observed that in the Full Trust condition, participants high in Honesty-Humility (above the median) were descriptively more likely to underreport the income-maximizing outcome ($P = 0.33$ [0.14; 0.51]) as compared to participants low in Honesty-Humility (below the median; $P = 0.11$ [0; 0.31]; $\Delta G^2(1) = 2.73$, $p = .10$).

However, in Studies 2 and 3, we did not manage to obtain similar results (see the “Confirmatory analyses” sections of Studies 2 and 3). After visually examining the data (see Figures 2 and 3), we concluded that although in Study 2 there was no underreporting of the income-maximizing outcome even among participants with the highest scores in Honesty-Humility, there was a small proportion of individuals who may have underreported this outcome in Study 3. The former is indicated by the 95% CI of the probability slope not falling below the expected probability of reporting the income maximizing outcome assuming full honesty (see Figure 2), while the latter is indicated by the entire 95% confidence interval of the probability slope falling entirely below the expected probability of reporting the income maximizing outcome assuming full honesty for the highest scores in Honesty-Humility (see Figure 3). Following this observation, we estimated that the lowest value of Honesty-Humility above which the 95% confidence interval of the probability in question falls below the expected probability (assuming full honesty), equals approximately 4.41 ($P = 0.15$; 95% CI [0.12; 0.16]). This indicates that for the participants with Honesty-Humility scores above or equal to 4.41 (which corresponds to

approximately the top 10%)¹, the probability of reporting the income-maximizing outcome was lower than expected assuming full honesty. Because this proportion was much smaller than what was observed in Study 1 (where the probability in question was lower than expected for the top 67.37%; see Figure 3), we decided to apply the 10% threshold across all three studies. This allows us to make rather conservative conclusions about trustworthy dishonesty only with regard to the participants with the very highest scores in Honesty-Humility (rather than to the majority of participants, as it would be indicated in Study 1).

¹ More specifically, it corresponds to 9.40%. However, we assume the value of 10% for simplicity.

Figure 1. Probability of reporting the income maximizing outcome (six) by Honesty-Humility in Study 1. The dashed horizontal line indicates the expected probability of reporting the income maximizing outcome assuming full honesty. The gray ribbon illustrates the 95% confidence interval. The dotted vertical line indicates the lowest value of Honesty-Humility above which the 95% CI of the probability slope falls below the expected probability of reporting the income maximizing outcome assuming full honesty (3.3; $P = 0.14$; 95% CI [0.12; 0.165]). The value of Honesty-Humility is larger or equal to 3.3 for 67.37% of the participants. $N = 1,713$.

Probability of reporting a six by Honesty-Humility

Figure 2. Probability of reporting the income maximizing outcome (six) by Honesty-Humility in Study 2. The dashed horizontal line indicates the expected probability of reporting the income maximizing outcome assuming full honesty. The gray ribbon illustrates the 95% confidence interval. The 95% CI does not fall below the expected probability of reporting the income maximizing outcome assuming full honesty. $N = 2,230$

Figure 3. Probability of reporting the income maximizing outcome (six) by Honesty-Humility in Study 3. The dashed horizontal line indicates the expected probability of reporting the income maximizing outcome assuming full honesty. The gray ribbon illustrates the 95% confidence interval. The dotted vertical line indicates the lowest value of Honesty-Humility above which the 95% CI of the probability slope falls below the expected probability of reporting the income maximizing outcome assuming full honesty (4.41; $P = 0.15$, 95% CI [0.12, 0.16]). The value of Honesty-Humility is larger or equal to 4.41 for 9.40% of the participants. $N = 3,137$." (pp. 1-6 in the Supplemental Material)

In addition, as a response to your comment, we adapted our suggestion in the Discussion section such that it now reads:

“Relatedly, the majority of the conducted analyses were exploratory—especially with regard to which low and top percentages were applied to separate people low and high in Honesty-Humility—, pointing at a necessity to conduct pre-registered, confirmatory studies on this topic, based on the effect sizes and patterns found across our studies.” (p. 29)

Minor issues:

- 7. The manuscript contains several typos, e.g. starting in the Abstract “We found that, when offered full trust, participants high in Honesty-Humility (the top 10%)” and several unusual formulations, e.g. the word “trustworthily” or “the trustor offered no trust”. Careful proofread and English editing is recommended.’**

Response:

We carefully proofread the manuscript before resubmission.

- 8. Table 1 presents the payoffs for all studies using pounds. In the note it states that Study 2, used Dollars. Currently, it is not directly clear what amounts of Dollars ppts could win. Why not present the payoffs for all three studies next to each other to make it a bit clearer? This way it is also easier to grasp that the No trust condition was not included in Study 2 & 3. Also in the note you refer to ““High Trust” while in the columns to “Full Trust”.**

Response:

We now present the payoffs for all three studies next to each other in Table 1 (pp. 11-12). In addition, we consistently use the term “Full Trust” in the text, tables, and table notes.

- 9. In the General Discussion, the sentence “Specifically, we found that, when fully trusted, participants high in Honesty-Humility chose to lie about not obtaining the outcome that would maximize their own incentive at the cost of the trustor, and/or to lie that they obtained the outcome(s) that would equalize the incentives between themselves and their trustor” Is very complicated and hard to grasp. Consider rephrasing.**

Response:

We rephrased this sentence as following:

“Specifically, we found that, when fully trusted, some participants high in Honesty-Humility chose to lie about not obtaining the outcome that would maximize their own outcome at the cost of the trustor. In addition, when fully trusted, some participants high in Honesty-Humility lied that they obtained the outcome(s) that would equalize the incentives between themselves and their trustor.” (pp. 26-27)

References

- Abbink. (2002). An Experimental Bribery Game. *Journal of Law, Economics, and Organization*, *18*(2), 428–454. <https://doi.org/10.1093/jleo/18.2.428>
- Abbink. (2004). Staff rotation as an anti-corruption policy: An experimental study. *European Journal of Political Economy*, *20*(4), 887–906.
<https://doi.org/10.1016/j.ejpoleco.2003.10.008>
- Ashton, M. C., & Lee, K. (2007). Empirical, Theoretical, and Practical Advantages of the HEXACO Model of Personality Structure. *Personality and Social Psychology Review*, *11*(2), 150–166. <https://doi.org/10.1177/1088868306294907>
- Ashton, M. C., & Lee, K. (2008). The HEXACO Model of Personality Structure and the Importance of the H Factor. *Social and Personality Psychology Compass*, *2*(5), 1952–1962.
<https://doi.org/10.1111/j.1751-9004.2008.00134.x>
- Balliet, D., & Van Lange, P. A. M. (2013). Trust, conflict, and cooperation: A meta-analysis. *Psychological Bulletin*, *139*(5), 1090–1112. <https://doi.org/10.1037/a0030939>
- Bolton, G. E., & Ockenfels, A. (2000). ERC: A Theory of Equity, Reciprocity, and Competition. *American Economic Review*, *90*(1), 166–193. <https://doi.org/10.1257/aer.90.1.166>
- Diebels, K. J., Leary, M. R., & Chon, D. (2018). Individual Differences in Selfishness as a Major Dimension of Personality: A Reinterpretation of the Sixth Personality Factor. *Review of General Psychology*, *22*(4), 367–376. <https://doi.org/10.1037/gpr0000155>
- Gerlach, P., Teodorescu, K., & Hertwig, R. (2019). The truth about lies: A meta-analysis on dishonest behavior. *Psychological Bulletin*, *145*(1), 1–44.
<https://doi.org/10.1037/bul0000174>

- Gross, J., Leib, M., Offerman, T., & Shalvi, S. (2018). Ethical Free Riding: When Honest People Find Dishonest Partners. *Psychological Science, 29*(12), 1956–1968.
<https://doi.org/10.1177/0956797618796480>
- Heck, D. W., Thielmann, I., Moshagen, M., & Hilbig, B. E. (2018). Who lies? A large-scale reanalysis linking basic personality traits to unethical decision making. *Judgment and Decision Making, 13*(4), 356–371.
- Hunt, J. (2004). *Trust and Bribery: The Role of the Quid Pro Quo and the Link with Crime* (No. w10510; p. w10510). National Bureau of Economic Research.
<https://doi.org/10.3386/w10510>
- Jiang, T., Lindemans, J. W., & Bicchieri, C. (2015). Can Trust Facilitate Bribery? Experimental Evidence from China, Italy, Japan, and The Netherlands. *Social Cognition, 33*(5), 483–504. <https://doi.org/10.1521/soco.2015.33.5.483>
- Kelley, H. H., Holmes, J. G., Kerr, N. L., Reis, H. T., Rusbult, C. E., & Lange, P. A. M. V. (2003). *An Atlas of Interpersonal Situations*. Cambridge University Press.
- Knack, S., & Keefer, P. (1997). Does Social Capital Have an Economic Payoff? A Cross-Country Investigation. *The Quarterly Journal of Economics, 112*(4), 1251–1288.
<https://doi.org/10.1162/003355300555475>
- Köbis, N. C., van Prooijen, J.-W., Righetti, F., & Van Lange, P. A. M. (2016). Prospection in Individual and Interpersonal Corruption Dilemmas. *Review of General Psychology, 20*(1), 71–85. <https://doi.org/10.1037/gpr0000069>

- Köbis, N. C., van Prooijen, J.-W., Righetti, F., & Van Lange, P. A. M. (2017). The Road to Bribery and Corruption: Slippery Slope or Steep Cliff? *Psychological Science*, *28*(3), 297–306. <https://doi.org/10.1177/0956797616682026>
- Köbis, N. C., Verschuere, B., Bereby-Meyer, Y., Rand, D., & Shalvi, S. (2019). Intuitive Honesty Versus Dishonesty: Meta-Analytic Evidence. *Perspectives on Psychological Science*, *14*(5), 778–796. <https://doi.org/10.1177/1745691619851778>
- Lambsdorff, J. G., & Frank, B. (2011). Corrupt reciprocity – Experimental evidence on a men’s game. *International Review of Law and Economics*, *31*(2), 116–125. <https://doi.org/10.1016/j.irl.2011.04.002>
- Leib, M., Moran, S., & Shalvi, S. (2019). Dishonest helping and harming after (un)fair treatment. *Judgment and Decision Making*, *14*(4), 423–439.
- Levine, E. E., Bitterly, T. B., Cohen, T. R., & Schweitzer, M. E. (2018). Who is trustworthy? Predicting trustworthy intentions and behavior. *Journal of Personality and Social Psychology*, *115*(3), 468–494. <https://doi.org/10.1037/pspi0000136>
- Lilleholt, L., Schild, C., & Zettler, I. (2020). Not all computerized cheating tasks are equal: A comparison of computerized and non-computerized versions of a cheating task. *Journal of Economic Psychology*, *78*, 102270. <https://doi.org/10.1016/j.joep.2020.102270>
- Morris, S. D., & Klesner, J. L. (2010). Corruption and Trust: Theoretical Considerations and Evidence From Mexico. *Comparative Political Studies*, *43*(10), 1258–1285. <https://doi.org/10.1177/0010414010369072>
- Moshagen, M., & Hilbig, B. E. (2017). The statistical analysis of cheating paradigms. *Behavior Research Methods*, *49*(2), 724–732. <https://doi.org/10.3758/s13428-016-0729-x>

- Moshagen, M., Hilbig, B. E., & Zettler, I. (2018). The dark core of personality. *Psychological Review*, *125*(5), 656–688. <https://doi.org/10.1037/rev0000111>
- Porta, R. L., Lopez-de-Silanes, F., Shleifer, A., & Vishny, R. W. (1996). *Trust in Large Organizations* (Working Paper No. 5864). National Bureau of Economic Research. <https://doi.org/10.3386/w5864>
- Schild, C., Heck, D. W., Ścigala, K. A., & Zettler, I. (2019). Revisiting REVISE: (Re)Testing unique and combined effects of REminding, VIvisibility, and SElf-engagement manipulations on cheating behavior. *Journal of Economic Psychology*, *75*, 102161. <https://doi.org/10.1016/j.joep.2019.04.001>
- Ścigala, K. A., Schild, C., Heck, D. W., & Zettler, I. (2019). Who Deals With the Devil? Interdependence, Personality, and Corrupted Collaboration. *Social Psychological and Personality Science*, *10*(8), 1019–1027. <https://doi.org/10.1177/1948550618813419>
- Soraperra, I., Weisel, O., Zultan, R., Kochavi, S., Leib, M., Shalev, H., & Shalvi, S. (2017). The bad consequences of teamwork. *Economics Letters*, *160*, 12–15. <https://doi.org/10.1016/j.econlet.2017.08.011>
- Tangney, J. P., & Dearing, R. L. (2004). *Shame and guilt*. Guilford Press.
- Thielmann, I., & Hilbig, B. E. (2015a). Trust: An Integrative Review from a Person–Situation Perspective. *Review of General Psychology*, *19*(3), 249–277. <https://doi.org/10.1037/gpr0000046>
- Thielmann, I., & Hilbig, B. E. (2015b). The Traits One Can Trust: Dissecting Reciprocity and Kindness as Determinants of Trustworthy Behavior. *Personality and Social Psychology Bulletin*, *41*(11), 1523–1536. <https://doi.org/10.1177/0146167215600530>

- Thielmann, I., Spadaro, G., & Balliet, D. (2020). Personality and prosocial behavior: A theoretical framework and meta-analysis. *Psychological Bulletin*, *146*(1), 30–90.
<https://doi.org/10.1037/bul0000217>
- van Baar, J. M., Chang, L. J., & Sanfey, A. G. (2019). The computational and neural substrates of moral strategies in social decision-making. *Nature Communications*, *10*(1), 1483.
<https://doi.org/10.1038/s41467-019-09161-6>
- Van Lange, P. A. M., & Balliet, D. (2015). Interdependence theory. In M. Mikulincer, P. R. Shaver, J. A. Simpson, & J. F. Dovidio (Eds.), *APA handbook of personality and social psychology, Volume 3: Interpersonal relations*. (pp. 65–92). American Psychological Association.
<https://doi.org/10.1037/14344-003>
- Weisel, O., & Shalvi, S. (2015). The collaborative roots of corruption. *Proceedings of the National Academy of Sciences*, *112*(34), 10651–10656.
<https://doi.org/10.1073/pnas.1423035112>
- Wouda, J., Bijlstra, G., Frankenhuys, W. E., & Wigboldus, D. H. J. (2017). The Collaborative Roots of Corruption? A Replication of Weisel & Shalvi (2015). *Collabra: Psychology*, *3*(1), 27.
<https://doi.org/10.1525/collabra.97>
- Zettler, I., Thielmann, I., Hilbig, B. E., & Moshagen, M. (2019). The Nomological Net of the HEXACO Model of Personality: A Large-scale Meta-analytic Investigation. *Perspectives on Psychological Science*.